# Suicide capability within the ideation-to-action framework: A systematic scoping review

**Luke T. Bayliss** *, **Steven Christensen, Andrea Lamont-Mills, Carol du Plessis**

University of Southern Queensland, Queensland, Australia

* Luke.Bayliss@usq.edu.au

## Abstract

Suicide capability is theorised to facilitate the movement from suicidal ideation to suicide attempt. Three types of contributors are posited to comprise suicide capability: acquired, dispositional, and practical. Despite suicide capability being critical in the movement from ideation-to-attempt, there has been no systematic synthesis of empirical evidence relating to suicide capability that would enable further development and refinement of the concept. This study sought to address this synthesis gap. A scoping review was conducted on suicide capability studies published January 2005 to January 2022. Eleven electronic databases and grey literature sources were searched returning 5,212 potential studies. After exclusion criteria application, 90 studies were included for final analysis. Results synthesis followed a textual narrative approach allocating studies based on contributors of suicide capability. Most studies focused on investigating only one factor within contributors. Painful and pro-vocative events appear to contribute to acquired capability more so than fearlessness about death. Whilst emerging evidence for dispositional and practical contributors is promising, the small number of studies prevents further conclusions from being drawn. An unexpected additional cognitive contributor was identified. The focus of a single factor from most studies and the limited number of studies on contributors other than acquired capability limits the theoretical development and practical application of suicide capability knowledge. Given that suicide is a complex and multifaceted behaviour, future research that incorporates a combination of contributors is more likely to advance our understandings of suicide capability.

## Introduction

Globally, approximately 700,000 individuals die by suicide every year [1]. For every suicide, there are an estimated 20 to 40 attempts [2]. For every attempt, an additional one and half [3] to three [4] individuals are thought to experience suicidal ideation. Thus, not everyone who experiences suicidal ideation will necessarily attempt suicide and not every attempt will result in a death by suicide. However, given the potential lethal and non-lethal (e.g., permanent disability) consequences of an attempt, identifying and understanding factors that move someone from ideation to attempt is paramount.

**Competing interests:** The authors have declared that no competing interests exist.

Suicide research is increasingly being guided by the ideation-to-action framework [5]. The framework aims to understand the movement from thinking about suicide to attempting suicide based on the premise that suicidal ideation and suicide attempt are related but distinct behaviours [6, 7]. The framework theorises that a core component of the movement from ideation-to-action is an individual's capability for suicide, which is a combination of contributors that facilitate an individual to attempt suicide [8, 9]. Theories of suicidal behaviour that sit within the ideation-to-action framework have identified three core contributors that are argued to be involved in an individual's capability for suicide.

The first contributor, acquired capability, comes from the Interpersonal Theory of Suicide (IPTS) [10, 11]. Acquired capability refers to lowered fearlessness about death and an elevated tolerance of physical pain resulting from habituation to painful and/or provocative events (e.g., childhood maltreatment, combat exposure [11]) Accordingly, an individual with an increased acquired capability is more likely to move from suicidal thoughts to a lethal (or near lethal) suicide attempt [11]. The Integrated Motivational-Volitional Model [12, 13], expands upon acquired capability by adding distinct volitional factors that facilitate a suicide attempt. These include access to means, exposure to suicide, impulsivity, and mental imagery. A weakness of these two theories is that they conceptualise suicide capability as consisting of individual contributors and focus on these single contributors when attempting to understand the movement from ideation-to-action. Given suicide behaviour is multifaceted [14], it is important to consider contributors in combination as a single contributor is unlikely to capture the complexity that underpins the movement from ideation to attempt [15].

More recently suicide capability has been conceptualised within the Three-Step Theory of Suicide [9, 16], as consisting of three contributors: acquired [10, 11], dispositional, and practical. Dispositional contributors refer to genetics, temperaments, and personality factors that may increase or decrease capability [16]. For example, being born with low sensitivity to pain may increase suicide capability [9]. Practical contributors are those that increase knowledge of and access to lethal means [21]. An example is an individual who is well acquainted with firearms because they have both the experience with and access to a lethal means, thus increasing their capability for suicide. Contributors from the previous two models of suicide have been incorporated into these three overarching contributors of capability. The theory argues that an individual develops a capability for suicide through this combination of contributors [17].

Few literature reviews have been conducted to examine the empirical evidence base for suicide capability and most have focused on only one theory within the ideation-to-action framework, the IPTS. One systematic review found equivocal support for acquired capability with only half of the studies providing support for its predictive ability in relation to suicide attempt [18]. In addition to this, a meta-analysis identified weak relationships between acquired capability and suicide attempts [19]. The authors suggested that acquired capability may be more complex than first proposed with other components such as genetics likely contributing to capability. Thus suggesting that on its own acquired capability may not be sufficient to explain the movement from ideation to attempt. A recent pre-print narrative review concluded that there is evidence that both supports and contradicts suicide capability as conceptualised by the IPTS [20]. Despite acknowledging that an important area for research is whether suicide capability is necessary for suicidal behaviour to occur, the review did not focus on potential relationships between suicidal behaviour and suicide capability. It is critical to focus on these potential relationships for greater clarification of the role suicide capability has in the movement from ideation to attempt before considering the necessity of capability. A second narrative review provided a conceptual update on contributors of suicide capability, adding factors that had not been previously considered as associated with capability [21]. These were, personality traits (e.g., sensation seeking) and interoceptive impairments (e.g., insensitivity to

physical and emotional states), which were added to the dispositional contributor, with exposure to suicide being conceptualised as a practical contributor.

The "explosion of work in recent years" [21 p6] and recent theoretical developments of suicide capability within the Integrated Motivational and Volitional model [12, 13] and the Three-Step Theory of Suicide [9, 16] raise questions about the utility of previous reviews to guide future development of the construct. These reviews have either been narrow in their focus as is consistent with the aims of a systematic review (i.e., only focused on acquired capability) [18, 19], narrative (i.e., potentially overlooking all and/or novel contributors) [20–22], did not explore whether suicidal behaviours were related to suicide capability [20], and/or are dated (i.e., the last systematic review was published 5 years ago) and thus may not capture more recent research. Given the timing and focus of these reviews, they do not and/or could not capture studies that include other contributors of suicide capability as suggested by the Integrated Motivational and Volitional model [12, 13] and the Three-Step Theory [9, 16]. Not including contributors beyond acquired capability potentially prevents theoretical and practical application progress because it overlooks the complex and multifaceted nature of suicidal behaviours by reducing capability to a single linear construct. Without a more recent and comprehensive synthesis of the literature it will be difficult for the ideation-to-action framework to move forward in its understanding of what contributes to someone moving from thinking about taking their own life to doing so. Synthesising the literature on suicide capability allows for an evidential reference point containing commonalities and differences of findings and knowledge gaps to be identified. This reference point can then be used to guide future research with the aim of enhancing theoretical understandings of suicide capability that can be used to design intervention and prevention strategies.

Recent theoretical developments and past review limitations necessitate an up-to-date mapping of evidence relating to contributors of suicide capability and potential relationships with suicidal behaviours. A scoping review is useful for doing this as it maps the literature relating to a research area to identify key concepts and knowledge gaps to inform future research and practice [23]. The aim of this scoping review was to systematically capture, collate, and synthesise the empirical research that has been conducted on contributors of suicide capability within the ideation-to-action framework. It did not test the efficacy or predictability of contributors as a systematic review or meta-analysis would aim to do. Instead, this review focused on bringing to the forefront what has been found within and across contributors of suicide capability thereby facilitating the field to take stock of where suicide capability is at this point in time. Understanding the current state of the field will allow for the identification of gaps in knowledge that should be the focus of future research.

## Method

The protocol for this scoping review has previously been published [24] and was based on Arksey and O'Malley's [25] scoping review methodology, with Levac et al.'s [26] and Peters et al.'s [23] recommendations being adopted.

### Stage 1: Identifying the research question

Contributors of an individual's capability for suicide within the ideation-to-action framework, published or translated into English, comprised the review's concept and context. The population was adults aged 18 years or above who had attempted suicide. Children and adolescents were excluded from this review because, while children and adolescents do also attempt and die by suicide, there may be psychosocial factors that are unique to this population, such as underdeveloped emotional regulation [27] and coping skills [28] that could contribute to their

capability for suicide. Further, research suggests that children and adolescent suicide attempt motivations differ significantly from adults which is often in the context of interpersonal problems [29, 30]. Given these capability and motivation considerations, this population warrants its own review and thus children and adolescents were not included in this review.

Four questions guided this review. They were as follows:

1. What is known about suicide capability as conceptualised within the ideation-to-action framework?

2. What methods have been utilised?

3. What limitations have been identified?

4. What research gaps are present?

## Stage 2: Identifying relevant studies

On 14 December 2020, the first and second author (LTB and SC hereafter) independently conducted the search for relevant studies using the search strategy (i.e., suicid* AND attempt* AND capa* OR "access to means") as outlined in [24]. Search results were recorded in a Microsoft Excel spreadsheet. The Cochrane Database of Systematic Reviews, the Database of Abstracts of Reviews and Effects, the International Prospective Register of Systematic Reviews, and the Joanna Briggs Institute (JBI) Evidence Synthesis journal were first searched to identify any previous or prospective reviews on suicide capability. No reviews in addition to the reviews [18–22] mentioned previously were identified. Following this, the eleven electronic databases below were independently searched one-at-a-time in the following order:

• Academic Search Ultimate

• APA PsycArticles

• APA PsycInfo

• CINAHL

• Psychology and Behavioural Sciences

• Sociology Source Ultimate

• PubMed

• Science Direct

• Wiley Online

• Taylor and Francis

• ProQuest dissertations and theses.

The grey literature database (www.opengrey.eu), Google Scholar, and the webpages of suicide organisations from Australia, the United States of America (U.S.), and Europe were then searched by both LTB and SC. LTB also examined the reference lists of two narrative reviews (i.e., [21, 22]) on suicide capability for any missed studies.

## Stage 3: Study selection

Reflecting the iterative nature of scoping reviews [25], amendments were made throughout the study selection phase. LTB and SC independently removed duplicates using EndNote (V9.2)

**Table 1. Inclusion and exclusion criteria.**

| Inclusion criteria | Exclusion criteria |
|---|---|
| Participants aged 18 years or older (for studies comprising a mixture of ages, the mean age needed to be 18.0 years or older; modified from protocol) | Entire sample aged under 18 years |
| Sample included participants with a history of suicide attempt(s) [32] and/or death by suicide | Sample only included participants with a history of non-suicidal self-injury (NSSI), assisted suicide, or suicidal ideation |
| Direct measure of suicide capability (e.g., acquired capability for suicide scale) or measures hypothesised to contribute to suicide capability (e.g., pain tolerance) | No measure related to suicide capability or indicators of suicide capability |
| Research studies needed to include a result relating to suicide capability and suicide attempt or death by suicide. This could be results that compared groups (i.e., suicide attempters and/or deaths by suicide to controls and/or suicide ideators), or results that indicated how suicide capability was associated with suicide attempts and/or deaths by suicide through correlational or qualitative studies. | Suicide prevalence study, scale development or validation study, or editorial; modified from protocol |
| Published or translated to English | No English translation could be located |
| Published from January 2005 | Published prior to January 2005 |

prior to screening. Using the eligibility criteria in Table 1, LTB and SC independently screened titles and abstracts with remaining studies having their full texts assessed against the eligibility criteria. Corresponding authors from studies not published in English were emailed to request translations, however none were available. When there was uncertainty about study eligibility ($n$ = 32), the study was discussed against the eligibility criteria for inclusion or exclusion with consensus reached for all 32 studies. Reasons for excluding each study at each step were recorded in the Microsoft Excel spreadsheet.

## Stage 4: Data extraction

A pilot data charting template [31] was modified and initially used by LTB. According to Perry et al. [33], to ensure relevant information is being captured the template should be pretested. Thus, the template was discussed and pretested with SC after trialling with five studies. The template was then refined to include contributor measures to better address the second research question. Data was extracted independently by LTB and included study information (i.e., author(s), year, country, title), study aim/hypotheses, sample characteristics, study design, suicide attempt measure, contributing factor(s) measure, results, and study limitations. After data extraction was completed, a random number generator identified 20% of articles that were audited by SC to ensure consistency and accuracy of extraction. No data extraction errors were identified.

While scoping reviews do not typically assess the quality of studies included in a review [25], an aim of this review was to identify limitations within the suicide capability literature therefore each study was assessed for quality. This was completed by LTB using adapted JBI critical appraisal tools [see 34–37]. In accordance with JBI guidelines and after research team discussion, a point scoring system was allocated alongside the appraisal framework to assess study quality. For this study, LTB appraised each study by giving each item on the checklist either a 'yes', (one point) or a 'no' (zero points). Scores were then added and converted into a percentage. Similar to the Pyle et al. [38] scoring system, studies scoring greater than 80% were deemed high quality, studies scoring between 50% - 80% were medium quality, and low-quality were studies that scored less than 50%.

## Stage 5: Results synthesis

Synthesisation of results was conducted by LTB with descriptive results of the studies collated first. Following this, textual narrative synthesis was used as a stepwise approach to synthesise and analyse studies [39]. To do this, data extraction templates were uploaded into NVivo [40] with studies grouped together based on theoretical contributors of suicide capability [9, 16] to help structure the findings and present results. Studies that were not aligned with previously identified contributors were allocated to an unidentified group. Sub-factor groups were identified based on variables within each contributor. The second step involved the production of textual descriptions for each article and included what the study provided towards understanding suicide capability. Finally, similarities and differences were synthesised within sub-groups to draw conclusions about contributors of suicide capability within the ideation-to-action framework.

## Stage 6: Amended and updated search

Based on peer-review feedback of study findings, the search strategy was amended to: suicid* AND attempt* OR behaviour* OR behavior* AND capa* OR "access to means". The term "behavio[u]r*" was added to capture studies that use suicidal behaviour instead of suicide attempt. The search was re-run on 17 January 2022 with the same two reviewers (i.e., LTB and SC) independently conducting the search. Search results were recorded in the same Microsoft Excel file and imported to the existing EndNote (V9.2) library. Stages two through five were again completed as previously outlined but only on the newly identified studies.

# Results

As a result of the two search strategies, 5,212 articles were originally identified (1,715 from the first search and 3,497 from the second search). After screening process, 90 articles were included for final analysis as shown in Fig 1 [41].

## Study methodologies

Most studies were peer-reviewed (*n* = 85) with four dissertations and one grey literature source also being included. Despite calls for qualitative suicidology research [42, 43], most studies were quantitative (*n* = 81) and cross-sectional (*n* = 57) in nature with only nine qualitative studies being identified [44–52]. In terms of quality, 6 studies were appraised as low quality, 45 medium, and 39 as high. Low quality study issues centred on measures that were psychometrically questionable (e.g., using incomplete measures or individual items from measures), and/or confounding factors (e.g., gender) not being identified or no mention of statistical strategies (e.g., matching of participants [53]) being used to deal with those factors. These same confounding factors were also common in quantitative medium quality studies, but statistical strategies were included to deal with those factors thus differentiating low and medium quality studies. For qualitative studies, the main quality concerns centred on there being no evidence or statement addressing the researcher's cultural/theoretical orientation and/or the influence of the researcher on the research and vice versa.

How suicide and suicide attempts were measured is displayed in Table 2. Death by suicide was consistently measured using coroner/medical examiner reports that often- informed government statistics (e.g., Web-based Injury Statistics Query and Reporting System [54]), and suicide registers [55]. Suicide attempts, however, were measured inconsistently. Most studies (*n* = 21) asked the participant one question about whether they had attempted suicide with questions differing in linguistic structure (see [56, 57]). Ten studies assessed participant suicide

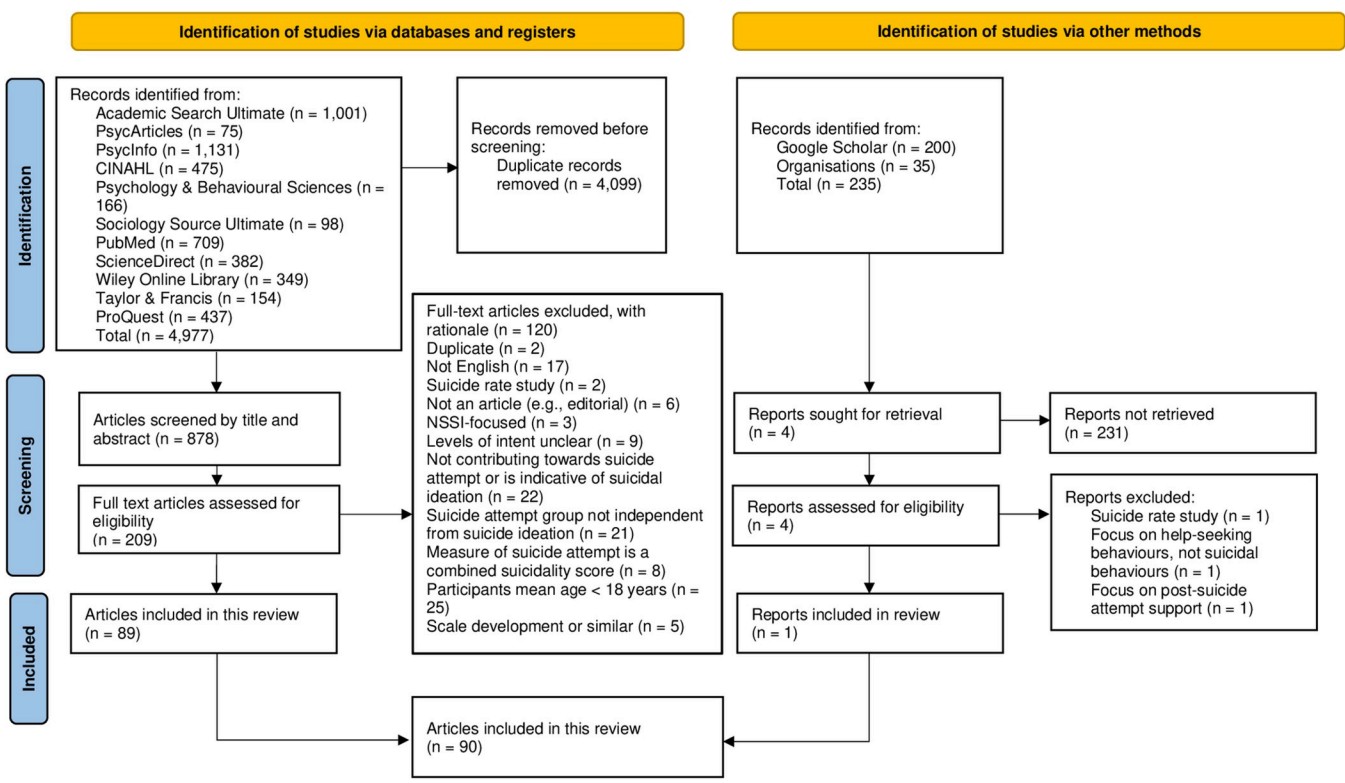

**Fig 1. PRISMA flow diagram of articles election process [41].**

attempt using an interview conducted by a psychiatrist/medical professional. The remaining 59 studies used 16 different unpublished and published measures (e.g., Lifetime Parasuicide Count (LPC); Linehan & Comtois, 1996, as cited in [58]); Suicide Behavior Questionnaire–Revised (SBQ-R) [59]) or used admission to a hospital/medical centre because of a suicide attempt as the attempt measure.

Diversity was also evident when measuring contributors of suicide capability (see Table 2). Some consistency was evident in the use of either the Acquired Capability for Suicide Scale (ACSS) [60] (*n* = 8) or the Acquired Capability for Suicide Scale–Fearlessness About Death (ACSS-FAD) [61] (*n* = 8) to measure acquired capability. However, five studies used either a small number of items or a single item from the ACSS that the researcher/s had selected, or they used the short form of the ACSS. For example, Chu et al. [62] used a four-item version of the ACSS, Smith et al. [63] used an eight-item version, Blankenship [64] used the single ACSS item "I am not at all afraid to die", and Wolford-Clevenger et al. [65] used the short form of the ACSS.

Despite there being a Painful and Provocative Events Scale [60], most studies used measures that reflected the painful provocative event that was the focus of the study (e.g., trauma, NSSI, substance use). The Suicide Capacity Scale (SCS-3) [9, 66], which measures suicide capability as a multifactor concept was seldom used (see [67–69]), as most studies focused on single aspects of suicide capability (e.g., painful and provocative events). Of the nine qualitative studies, two used life charting [48, 51], two used thematic analysis [45, 46] and the remaining five studies used different qualitative approaches reflecting the heterogeneity of qualitative methodologies [70].

**Table 2. Study methodologies and appraisal.**

| Reference | Design | Suicide attempt measure | Contributing factor measure(s) | Quality appraisal |
|---|---|---|---|---|
| Abdollahpour Ranjbar et al. [135] | Cross-sectional | Clinical interview (i.e., Have you ever tried to commit suicide and if yes, then how many times, in your whole life have you tried to kill yourself?) | Stroop Test [174]. Wisconsin Card Sorting Test [175]. Raven's standard progressive matrices [176]. Cognitive emotion regulation questionnaire [177]. | High |
| Allbaugh et al. [85] | Cross-sectional | Reported to hospital for suicide attempt | ACSS [59]. Childhood Trauma Questionnaire [178]. | High |
| Ammerman et al. [119] | Cross-sectional | Suicide Behavior Questionnaire—Revised [58] | Form and Function of Self-Injury [179] | Medium |
| Anestis & Joiner [111] | Cross-sectional | Beck Scale for Suicidal Ideation [163] | A 6-item ACSS [59]. The Negative Urgency subscale from The Urgency, (lack of) Premeditation, (lack of) Perseverance, Sensation Seeking Impulsive Behavior Scale [119]. | Medium |
| Anestis et al. [112] | Cross-sectional | Beck Scale for Suicidal Ideation [163] | A 5-item ACSS [59]. The Negative Urgency subscale from The Urgency, (lack of) Premeditation, (lack of) Perseverance, Sensation Seeking Impulsive Behavior Scale [119]. Impulsive Behavior Scale [Rossotto, et al., as cited in 121]. | High |
| Anestis et al. [58] | Cross-sectional | Lifetime Parasuicide Count (Linehan & Comtois, 1996, as cited in [57]) | The Paced Auditory Serial Addition Task–Computerized Version [180] | High |
| Anestis et al. [106] | Cross-sectional | Beck Scale for Suicidal Ideation [163] | ACSS–FAD [60] | Medium |
| Anestis et al. [133] | Ecological | Web-based Injury Statistics Query and Reporting System [53] | Universal background checks refer to a requirement that individuals selling a gun use a local, state, or federal system (variable by state) to search for records indicating that the individual attempting to buy the gun is barred from doing so. Mandatory waiting periods refer to the amount of time required to pass between the purchase of a gun and the physical transfer of the weapon from the seller to the purchaser. | High |
| Aschrafi et al. [142] | Case control | Medical and coroner records | Luciferase activity in relative light units | Medium |
| Baer et al. [99] | Cross-sectional | "Over the course of your entire life, how many times have you intentionally harmed yourself with at least some intention of causing your own death?" | ACSS [59]. The Drug Use Questionnaire [Hien & First, 1991, as cited in 115]. | High |
| Baertschi et al. [104] | Cross-sectional | Admitted to psychiatric emergency department for suicide attempt | 5-items from the German Capability for Suicide Questionnaire [181] | Medium |
| Ben-Efraim et al. [121] | Case control | Medical Damage Rating Scale [164] | Single-nucleotide polymorphisms rs4792887 and rs16940665 in CRHR1 gene. Life Events section of the European Parasuicide Study Interview Schedule (Kerhof et al., 1989, as cited in [125]) version 5.1. The post-traumatic stress disorder (K) section of the Composite International Diagnostic Interview [182] version 2.1. | High |
| Beyond Blue [44] | Qualitative | Unclear, however interview data was used. | Lived experience | Medium |
| Biddle et al. [45] | Qualitative | Clinical records | Lived experience | Medium |
| Blankenship [64] | Cross-sectional | History of suicide attempt was assessed by asking the participant whether they attempted suicide according to Silverman et al. [30] definition, and, if so, the number of attempts they have made. | One item from the ACSS [59] "I am not at all afraid to die". The Trauma Experience Questionnaire was created for this study by combining and simplifying items from the Trauma History Questionnaire [183], the Life Events Checklist for DSM-5 Extended Version [184]. | Low |

(*Continued*)

**Table 2.** (Continued)

| Reference | Design | Suicide attempt measure | Contributing factor measure(s) | Quality appraisal |
|---|---|---|---|---|
| Brackman et al. [120] | Cross-sectional | The Suicidal Behaviors Questionnaire (Linehan, 1981, as cited in [120]) | ACSS-FAD [60]. Functional Assessment of Self-Mutilation (FASM; Lloyd et al., 1997, as cited in [120]). The BIOPAC stimulator module (STM100C BIOPAC Systems Inc., Santa Barbara, CA) was used with a BIOPAC stimulus isolation adapter and programmed to deliver brief shocks. | Low |
| Calati et al. [76] | Case control | Self-report and medical records | Single-nucleotide polymorphisms | Medium |
| Cao et al. [84] | Case-control | Death certification system | The Life Events Scale for the Elderly [185] | High |
| Carli et al. [130] | Cross-sectional | Study raters were specifically trained to discriminate between suicide attempters, ideators and self-mutilators. | Barratt Impulsivity Scale (7B version) [186] | High |
| Cheek et al. [72] | Cross-sectional | Depression section of the National Co-morbidity Survey-Replication [165] | Individuals who endorsed having used heroin, cocaine, and stimulants were asked an additional question asking whether they had used a needle to inject that particular substance in their lifetime. They were also asked if they had injected any other drug at least once. | High |
| Chelmardi et al. [67] | Cross-sectional | "How many times have you made an actual attempt to kill yourself in which you had at least some intent of death?" | The Depressive Symptom Index-Suicidality Subscale [187]. Suicide Plan: "Have you ever had a plan to kill yourself at a specific time (e.g. Monday), a specific place (e.g. home, street), or by a specific method (e.g. drug overdose), and did you have an alternative plan if your initial one faced unexpected problems?" Plutchik Impulsivity Scale [188]. Two items were used to find the likelihood of individual's exposure to family or friends' suicide or self-injurious behaviors. Self-Perceived Acquired Capability for Suicide [189]. Suicide Capacity Scale-3 [9, 66]. The Primary Care PTSD Screen for DSM-5 [190]. Three items were adapted from the non-suicidal self-injury assessment tool [191]. Two items with yes/no choices were adopted from the original Adverse Childhood Experiences Scale [192]. Suicidal Behaviors Questionnaire–Revised [58]. | High |
| Chu et al. [62] | Cross-sectional | Depressive Symptom Inventory: Suicidality Subscale [166] | An abbreviated four-item version of the ACSS [59] | Low |
| Chu et al. [77] | Cross-sectional | Self-Injurious Thoughts and Behaviors Interview [167] | Self-Injurious Thoughts and Behaviors Interview [167]. ACSS [59]. | High |
| Copeland et al. [89] | Cohort | Up to 16 years, Child and Adolescent Psychiatric Assessment [168]. From 19–30 years, Young Adult Psychiatric Assessment [169]. | Same as suicide attempt measures | High |
| Daruwala et al. [105] | Cross-sectional | Beck Scale for Suicidal Ideation [163] | The ACSS—FAD [60]. The sensation seeking subscale from The Urgency, (lack of) Premeditation, (lack of) Perseverance, Sensation Seeking Impulsive Behavior Scale [119]. The Liverpool Stoicism Scale [193]. The physical aggression and verbal aggression subscales from the Buss Perry Aggression Questionnaire [194]. | Medium |

(*Continued*)

**Table 2.** (Continued)

| Reference | Design | Suicide attempt measure | Contributing factor measure(s) | Quality appraisal |
|---|---|---|---|---|
| DeVille et al. [126] | Cross-sectional | Columbia Suicide Severity Rating Scale [170] | Multidimensional Assessment of Interoceptive Awareness [195].<br>Toronto Alexithymia Scale [196].<br>Behavioural tasks: breath-hold challenge, cold-pressor challenge, and heartbeat perception task. | High |
| Dhingra et al. [68] | Cross-sectional | Self-Injurious Thoughts and Behaviours Interview [167] | Suicide Capacity Scale [9, 66] | Medium |
| Duddin & Raynes [46] | Qualitative | British transport police fatality database | Suicide notes | High |
| Feltrin et al. [93] | Cross-sectional | National College Health Risk Behavior Survey (Franca & Colares, 2010, as cited in [93]) | The Lipp Adult Stress Symptom Inventory (Lipp, 2000, as cited in [96]) | Low |
| Forrest et al. [56] | Cross-sectional | "Have you ever made an actual attempt to kill yourself in which you had at least some intent to die?" | Interoceptive Awareness subscale of the Eating Disorder Inventory [197] | Medium |
| Govind [47] | Qualitative | Semi-structured interview that asked about suicide attempts, thoughts, gestures, and self-harm details. | Lived experience | High |
| Hardt et al. [82] | Cross-sectional | Lifetime suicidality was assessed by a question with five possible answers: (1) Suicide attempt, (2) Plan, (3) Ideation, (4) No suicidality, and (5) Don't know/ Refuse to answer. | Four questions concerned own physical abuse: (1) regular harsh punishment, (2) having been beaten so that bruises occurred, (3) parents' threatening behaviour, and (4) violence between parents.<br>Sexual abuse was assessed by posing the following three questions: (1) Did you have any unwanted sexual experience with someone at least five years older than you before you reached the age of 15? (2) If so, would you consider it as abuse? and (3) Who was the perpetrator? | Medium |
| Heiden-Rootes et al. [97] | Cross-sectional | During the past 12 months, did you try to kill yourself? | Respondents were asked, "Did any professional (such as a psychologist, counsellor, or religious advisor) try to make you identify only with your sex assigned at birth (in other words, try to stop you from being trans)?" Those answering "no" were categorized as no exposure to gender identity change efforts and those answering "yes" as having had exposure to gender identity change efforts at some point in their lifetime. A follow-up question among those with at least one exposure to gender identity change efforts asked, "Was this person a religious or spiritual counsellor/ advisor?," where a "yes" response further categorized participants into experiencing GICE within a religious setting. | High |
| Hsiao et al. [139] | Case-report | Admitted to psychiatry service at a medical centre after attempting suicide | The Autobiographical Memory Interview [198].<br>Aspects of Identity Questionnaire-IV [199].<br>Twenty Statements Test measure of identity [200]. | High |
| Huang et al. [117] | Cross-sectional | Modified Self-Injurious Thoughts and Behaviors Interview [167] | ACSS—FAD [60] | High |
| Joiner et al. [115] | Cross-sectional | Interviewer-rated form that included assessment about recent suicide attempt and lifetime suicide attempt history | Lifetime number of suicide attempts | High |
| Jordan et al. [98] | Cross-sectional | Admitted to hospital after suicide attempt | A modified version of the Violent Victimization scale from The MacArthur study of mental disorder and violence [201] and other life events that reflect painful and provocative events, such as threatening and/or assaulting others with a weapon and NSSI, were combined to give an overall score. | Medium |
| Jordan & Samuelson [101] | Cross-sectional | "I made a serious attempt to kill myself and it was only luck that I did not succeed" | Events in which the individual purposefully or accidentally injured or killed another were labelled "committing violence" | High |

*(Continued)*

**Table 2.** (*Continued*)

| Reference | Design | Suicide attempt measure | Contributing factor measure(s) | Quality appraisal |
|---|---|---|---|---|
| Jovičić et al. [127] | Cross-sectional | Information on suicide attempts was confirmed after inspection of official documents and patients' medical history. | Temperament Evaluation of Memphis, Pisa, Paris, and San Diego Autoquestionnaire [202]. The current Serbian version comprised 41 true/false items grouped into six temperament [203]. Big Five Plus 2 Personality Questionnaire, short version [204]. | Medium |
| Kasen et al. [128] | Cohort | "Did you (your child) ever try to kill yourself (him/herself)?" Youths and mothers responded to parallel interview items about suicide attempts by the youth. | A 7-item measure of impulsivity comprised of items adapted from established measures from [119] | Medium |
| Kene [78] | Cross-sectional | Patient medical charts | ACSS [59] | Medium |
| Kerbrat et al. [86] | Cross-sectional | Suicide Attempt Self-Injury Count interview [171] | ACSS [59]. ACSS–FAD [60]. "Please indicate the number of combat deployments during your entire military career". Suicide Attempt Self-Injury Count [171]. | High |
| Khazem & Anestis [17] | Cross-sectional | Self-Injurious Thoughts and Behaviors Interview [167] | Painful and Provocative Events Scale [59]. ACSS–FAD [60]. | High |
| Kishikawa et al. [140] | Case study—comparative | Death by suicide | Mini-Mental State Examination [as cited in 141] | Medium |
| Klonsky et al. [116] | Cross-sectional | Youth Risk Behavior Survey [172]. National Comorbidity Survey [165]. | "In your lifetime, how often have you intentionally hurt yourself—for example, by scratching, cutting, or burning—even though you were not trying to commit suicide?" The Urgency, (lack of) Premeditation, (lack of) Perseverance, Sensation Seeking Impulsive Behavior Scale [119]. The Inventory of Statements About Self-injury [119, 139] | High |
| Knowles et al. [124] | Cohort | Mini-International Neuropsychiatric Interview [173] | Cholesterol efflux capacity | High |
| Koweszko et al. [144] | Cross-sectional | Columbia Suicide Severity Rating Scale [170] | Oxidative stress components | High |
| Kunde et al. [48] | Qualitative | Queensland Suicide Register [54]. State Coroner's Court of New South Wales. | Psychology autopsy interviews were conducted with a close relative of the male farmer who died by suicide. | High |
| Law & Anestis [103] | Cross-sectional | Self-Injurious Thoughts and Behaviours Interview [167] | The Positive and Negative Affect Schedule [205]. An adapted version of the Pitman Protocol [206]. Adaption of the rumination induction protocol developed by Nolen-Hoeksema and Morrow [207]. S2 (in addition to S1 measures) Changes in Heart Rate (HR) derived from electrocardiogram (ECG) acquired using the Biopac MP150 Data Acquisition System and the BN-RSPEC wireless transmitters and receivers. | High |
| Law et al. [132] | Ecological | Suicides by jumping data from the Queensland Suicide Register [54] | Fencing barriers on bridge | Medium |
| Law et al. [118] | Cross-sectional | Lifetime Suicide Attempt Self-Injury Interview (Linehan & Comtois, 1996, as cited in [118]) | Pain tolerance and pain threshold were measured using a Wagner FPIX 25 pressure algometer. Distress Tolerance Test [208]. The Deliberate Self Harm Inventory [209]. | High |
| Leira et al. [71] | Case-control design in a naturalistic setting | Suicide data were gathered from the Norwegian Cause of Death Registry | Hospitalised from self-harm data were gathered from hospital records | High |
| Li et al. [122] | Case control | Interviews supervised and reviewed by a member of the research team and medical records when possible. | Single-nucleotide polymorphisms rs300774, rs7296262, and rs10437629 | Medium |

(*Continued*)

**Table 2.** (Continued)

| Reference | Design | Suicide attempt measure | Contributing factor measure(s) | Quality appraisal |
|---|---|---|---|---|
| Liu [74] | Cohort | "During the past 12 months, how many times did you actually attempt suicide?" | Dichotomous variable indicating whether either a family member or a friend had ever attempted suicide | Medium |
| Love & Durtschi [73] | Cohort | "During the past 12 months, how many times did you attempt suicide?" | Acquired capability was assessed by asking about any previous childhood abuse and previous suicide attempts | Medium |
| Martin [52] | N-of-1 case study | Death by suicide | Family suicides. Injuries. | Medium |
| McCarthy et al. [138] | Cross-sectional | The Mini International Neuropsychiatric Interview Suicidal Scale [173]. To provide further context, demographic and offence data were also collected from both self-report and case note review. | The Autism Quotient [210]. The Learning Disability Screening Questionnaire [211]. The Adult Self-Report Screen for ADHD: World Health Organisation [212]. The Mini International Neuropsychiatric Interview [173]. | Medium |
| Medeiros et al. [134] | Cross-sectional | Clinical record and an interview to confirm and clarify the record of the suicide attempt | Stroop Test [174]. Computerised version of the Wisconsin Card Sorting Test [175]. A modified Iowa Gambling Task [145]. | Medium |
| Miller [114] | Cross-sectional | Dataset is a collection of interviews with suicide attempt survivors. | Acquired capability was coded if there was a history of behaviours that would reduce fear of death or that would increase pain tolerance | Medium |
| Oakes-Rogers, & Slade [92] | Case series | Death had been deemed intentional by an independent Coroner | Prison and Probation Ombudsmen's independent reports on deaths in custody | Medium |
| Olié et al. [137] | Cross-sectional | Assessed at bipolar expert centres. A suicide attempt was defined as a self-damaging act carried out with some intent to die. | Wechsler Adult Intelligence Scale-III [213]. Stroop Colour–Word Interference Test reading parts [214]. Trail Making Test Part A [215]. The WAIS-III Processing Speed Index [216]. Verbal learning and memory was measured by the list-learning task of the California Verbal Learning Test [217]. The verbal fluency protocol [218]. Executive functioning was assessed with verbal fluency, the Trail Making Test part B and Stroop interference part (part 3). | High |
| Oshnokhah et al. [143] | Case control | Admittance to the emergency room after suicide attempt | 10 ml of blood samples. Ferric Ion Reducing Antioxidant Power (FRAP) Assay was used to measure total antioxidant capacity (NaxiferTM Kit). Lipid peroxidation was evaluated by measuring the amount of MDA in serum samples using Nalondi Kit ™. Superoxide Dismutase activity was measured through pyrogallol autoxidation. | Low |
| Pelton et al. [113] | Cross-sectional | Suicide Behaviours Questionnaire-Revised [58] | ACSS—FAD [60]. Vulnerability Experience Quotient [219]. | Medium |
| Pettit et al. [88] | Cross-sectional | Participants were asked if a suicide attempt had precipitated their entering treatment through a self-report psychosocial questionnaire. | The Life Experiences Survey [220] | High |
| Pisetsky et al. [81] | Cross-sectional | "Have you ever made an actual attempt to kill yourself in which you had at least some intent to die?" | Painful and Provocative Events Scale [59]. ACSS—FAD [60]. | Medium |
| Pitman et al. [131] | Case series | As recorded by the National Confidential Inquiry into Suicide and Safety in Mental Health which is a database that records deaths by suicide for people under the care of mental health service providers across the United Kingdom (U.K.) | Coded by psychiatrist | High |

*(Continued)*

**Table 2.** (Continued)

| Reference | Design | Suicide attempt measure | Contributing factor measure(s) | Quality appraisal |
|---|---|---|---|---|
| Price [80] | Case control | Suicide death reviews were provided by the California Department of Corrections and Rehabilitation to the Department of State Hospitals-Vacaville | Chronic, Acute, and Idiosyncratic inventory (Department of State Hospitals-Vacaville, as cited in [79]) | Medium |
| Rappaport et al. [75] | Cross-sectional | "Did you attempt suicide?" | Neuroticism subscale from the Eysenck Personality Questionnaire [221] | High |
| Raubenheimer & Jenkins [49] | Qualitative | Doctor on duty at emergency centre | Lived experience | Low |
| Richard-Devantoy et al. [136] | Case control | Suicide attempt history was verified by a psychiatrist, using an interview, medical records, and information from family or acquaintances. | Reading with Distraction Task [222]. Trail Making Test [223]. Rule Shift Cards [223]. The Go/No-Go test [223]. Baddeley Dual-Task Performance [223]. The Verbal Fluency Test [223]. The Stroop Color Test [223]. Hayling Sentence Completion test [224]. | High |
| Rogers et al. [57] | Cross-sectional | "Have you ever made a suicide attempt with at least some intent to die?" and number of past suicide attempts "How many times have you attempted suicide with at least some intent to die?" | ACSS [59]. Exercise Dependence Scale [225]. | Medium |
| Ryan et al. [83] | Cross-sectional | "Have you ever, at any point in your life, attempted taking your own life?" | Parent-initiated efforts to change youths' sexual orientation The first item asked: "Between ages 13 and 19, how often did any of your parents/caregivers try to change your sexual orientation (i.e., to make you straight)?" A second item asked: "Between ages 13 and 19, how often did any of your parents/caregivers take you to a therapist or religious leader to cure, treat, or change your sexual orientation?" | Medium |
| Shelef et al. [94] | Cross-sectional | Unknown—Suicide attempt was defined as: "A potentially self-injurious behavior, associated with at least some intent to die, as a result of the act. Evidence that the individual intended to kill him/herself, at least to some degree, can be explicit or inferred from the behavior or circumstance". | ACSS [59]. Perceived Stress Scale [226]. Perceived Army Stress Scale [227]. The Body Image Aberration scale [228]. | Medium |
| Shelef et al. [87] | Cross-sectional | Informed by mental health officer | ACSS [59]. Perceived Stress Scale [226]. | Medium |
| Shim et al. [109] | Cross-sectional | "I have made attempts to kill myself in the past" | The Korean version [109] of ACSS [59] | Medium |
| Smith et al. [90] | Cross-sectional | Suicide attempters reported at least one suicide attempt defined as per [30] | ACSS [59]. Combined Painful and Provocative Events Scale and Impulsive Behaviors Scale [86]. Life Experiences Survey [220]. | High |
| Smith et al. (Study 1) [63] | Cross-sectional | "Have you made any suicide attempts?" | The Eating Disorder Examination Questionnaire-4 [229] | Medium |
| (Study 2) [63] | Cross-sectional | Lifetime suicide attempts were measured by asking participants the number of times they had attempted suicide | An abbreviated eight-item version of the ACSS [59] | Medium |
| Smith et al. [102] | Cross-sectional | Suicide Attempt Self-Injury Interview [171] | ACSS and Painful and Provocative Events Scale [59]. Life Experiences Survey [220]. | High |
| Sokolowski et al. [79] | Case control | Medical Damage Rating Scale [164] | Single-nucleotide polymorphisms ($n$ = 113). ODC1 gene (Single-nucleotide polymorphisms rs1049500, rs2302614, and rs7559979) and the glutamatergic GRIN2B gene (through Single-nucleotide polymorphisms rs2268115 and rs220557). | High |

(*Continued*)

**Table 2.** (Continued)

| Reference | Design | Suicide attempt measure | Contributing factor measure(s) | Quality appraisal |
|---|---|---|---|---|
| Stoliker & Abderhalden [100] | Cross-sectional | "Have you ever attempted suicide?" | The CAGE-questionnaire [230]. Drug use (yes/no) was based on the aggregation of two survey items, which asked whether respondents had "used drugs other than those required for medical reasons" or "used prescription drugs other than what they are prescribed for" in the 12 months prior to current incarceration. Aggression, respondents were asked if they had gotten into fights when under the influence of drugs (yes/no) —that is, interpersonal violence while intoxicated. Respondents were further assessed according to whether they had engaged in self-harm while in jail before (yes/no) and if they had no wish to live (agree/disagree). Social support (yes/no) was based on the aggregation of two survey items, which asked whether respondents "received emotional support from friends/family" and if they "are satisfied with the level of support from friends/family." Loneliness (yes/no) was assessed according to the statement, "I feel lonely." The survey also captured whether respondents get enough sleep at night (yes/no). | Medium |
| Sunnqvist et al. [51] | Qualitative | Admitted to hospital after a suicide attempt | COPE-Inventory [231] | Medium |
| Suto & Arnaut [50] | Qualitative | "What did you do to harm yourself?" | A semi-structured interview with questions developed to gain an understanding of factors leading up to participant's suicide attempt(s) | Medium |
| Tull et al. [123] | Case control | The suicidality portion of the Mini International Neuropsychiatric Interview, Version 6.0 [171]. | On average, each [saliva] sample resulted in 3.5ug of total DNA or ~100 ng/ul. All samples were normalized to 50 ng/ul. Taqman genotyping was performed using pre-designed Taqman SNP Genotyping Assay for COMT Val158Met (rs4680, Cat#4362691). | High |
| Van Orden et al. [60] | Cross-sectional | Beck Scale for Suicide Ideation [169] | Painful and Provocative Events Scale and Impulsive Behaviors Scale [59]. 5-item ACSS [59]. | High |
| Van Orden et al. [91] | Case control | Cases were suicide decedents consecutively identified by the Chief Medical Examiners of Monroe and Onondaga, NY counties | Painful and provocative events operationalised as: previous number of suicide attempts, general aggression [232], relative died by suicide, and owned firearm in month prior to suicide | High |
| Wolford-Clevenger et al. [108] | Cross-sectional | Asking the participant whether they attempted suicide | ACSS [59]. Drug Use Disorders Identification Test [233]. Alcohol Use Disorder Identification Test [234]. | Medium |
| Wolford-Clevenger et al. [65] | Cross-sectional | Suicide Attempt Self-Injury Interview [171] | ACSS–Short Form [59]. Physical Violence Perpetration and Victimization subscales of the Revised Conflict Tactics Scale [235]. | Medium |
| Wong et al. [129] | Case control | Death by suicide–Coroner report | Impulsivity Rating Scale [236] | Medium |
| Yang et al. [69] | Cross-sectional | "Have you ever attempted to kill yourself in your life?" | Suicide Capacity Scale [9, 66]. The SCS was translated into Chinese and back translated to check for accuracy. The Chinese version of the ACSS-FAD was used (Li, 2014, as cited in [69]). | Medium |

(*Continued*)

**Table 2.** (Continued)

| Reference | Design | Suicide attempt measure | Contributing factor measure(s) | Quality appraisal |
|---|---|---|---|---|
| Zhao et al. [95] | Cross-sectional | Admitted to emergency room after a suspected suicide attempt, after which researchers conducted an interview to confirm the attempt. | Life events that resulted in psychological distress over the prior year were assessed using a 60-item scale developed specifically for use with suicidal individuals in China [237]. Interviewers and researchers reviewed all the material obtained from the two interviews and made a determination for each case of the relative importance of seven main causes of suicide (derived from previous studies of fatal and nonfatal suicide): family conflict, economic problems, low mood, alcohol or other substance abuse problems, other psychiatric conditions, physical illness, or other causes. | Medium |

**Sample characteristics.** Sample sizes and characteristics varied across studies. As shown in Table 3, more than half of all studies were conducted in the United States (U.S.), by U.S. based researchers. Sample sizes displayed in Table 4 highlight differences in sample sizes, ranging from an N-of-1 case study [52], to 64,770 participants within a population-based cohort study [71]. For ease of reporting, sample sizes have been grouped into four size ranges: five studies comprised more than 10,000 participants, 28 studies had a sample size between 501–10,000, another 31 had a sample in the 101–500 range, with the remaining 26 studies ranging in size from 1–100. Studies with larger samples often used community health datasets not primarily designed for suicide research as data (see [71–75]). Only seven studies reported conducting a priori power analyses to determine required sample (see [62, 64, 65, 76–79]).

As seen in Table 4, most studies focused on or had more suicide ideators than suicide attempters as the participants. On average, 26.77% of participants were either suicide attempters or individuals who died by suicide identified across studies that compared with suicide ideators and/or controls, but within studies the percentage of suicide attempters/death by suicide ranged from 0.14% [71] to 71.43% [80]. The mean age across all samples was 33.65 years ($SD$ = 9.20) (see Table 4). There were more females ($n$ = 77,277) than males ($n$ = 67,496) represented across studies. Nine studies included genders other than male or female. There were 21 single gender studies and when more than one gender was represented, most studies reported gender disproportionate samples, such as 91.9% males [62] and 93.90% females [81]. Seventeen studies had a gender difference split of less than 10% (i.e., less than a 45%/55% split).

Most study participants were White/Caucasian, with participants identifying as African American/Black, Latino/Hispanic, Bi/Multiracial, Asian, and Native Indian being less prevalent (refer to Table 4). Most studies focused on individuals without a psychiatric diagnosis ($n$ = 60), however of the 30 that did focus on psychiatric diagnosis, diagnoses included major depressive disorder, bipolar disorder, borderline personality disorder, and other disorder diagnoses such as eating, personality, anxiety, and schizophrenia spectrum (see Table 4). Participants came from various populations including university students ($n$ = 9), serving and veteran military personnel ($n$ = 8), incarcerated individuals ($n$ = 7), and interpersonal violence victims and perpetrators ($n$ = 2).

**Theoretical foundations.** Most studies that sought to theoretically ground their work drew upon the IPTS to do so [10, 11] (see S1 Table). However, over a third of studies were atheoretical ($n$ = 38). These studies typically sought to identify differences between suicide ideators and suicide attempters, often finding such differences. By not explicitly grounding their work in a theoretical model, it is unclear how these findings contribute to the development of theories.

**Table 3. Country of origin of grouped by geographic region.**

| Location | Number of studies per nationality of first author | Number of studies per nationality of sample; [76, 140] multiple nationalities |
|---|---|---|
| East Asia and Pacific | | |
| Australia | 3 | 3 |
| China | 3 | 4 |
| Hong Kong | 1 | 1 |
| Japan | 1 | 1 |
| New Zealand | 1 | 0 |
| Japan | 1 | 1 |
| Korea | 1 | 1 |
| New Zealand | 1 | 0 |
| Europe and Central Asia | | |
| France | 1 | 2 |
| Germany | 1 | 2 |
| Hungary | 0 | 1 |
| Italy | 2 | 2 |
| Netherlands | 1 | 0 |
| Norway | 1 | 1 |
| Poland | 1 | 2 |
| Portugal | 1 | 1 |
| Serbia | 1 | 1 |
| Switzerland | 1 | 1 |
| Sweden | 3 | 1 |
| Turkey | 1 | 0 |
| Ukraine | 0 | 2 |
| United Kingdom | 6 | 7 |
| Latin America and Caribbean | | |
| Brazil | 1 | 1 |
| Middle East and North Africa | | |
| Israel | 2 | 2 |
| Iran | 2 | 3 |
| North America | | |
| Canada | 2 | 0 |
| United States | 51 | 51 |
| Sub-Saharan Africa | | |
| South Africa | 2 | 2 |
| International | 0 | 1 |

## Contributors of suicide capability findings

Studies in this review aimed to understand contributors of suicide capability in three ways. One, quantitative studies sought to either compare suicide attempters or individuals who died by suicide, to suicide ideators and/or controls ($n = 31$), or two, were correlational studies that aimed to identify potential relationships between contributors of suicide capability with single group designs comprising suicide attempters or individuals who died by suicide ($n = 10$) or with multiple groups that also included controls and/or suicide ideators ($n = 40$). Three, all

**Table 4. Sample characteristics.**

| Reference | Sample size | Mean age (years) | Standard deviation (years) | Percentage of sample attempters/ suicides | Population | Male/Female/ Other % | Ethnicities[1] |
|---|---|---|---|---|---|---|---|
| Abdollahpour Ranjbar et al. [135] | 75 | 36.29 | 8.93 | 33.33% | Community—females | 0/100 | Iranian sample |
| Allbaugh et al. [85] | 179 | 36.35 | 10.55 | 100% | Clinical–hospitalised females from suicide attempt | 0/100 | 100% African American |
| Ammerman et al. [119] | 997 | 20.64 | 2.88 | 12.84% | U.S. university students with a history of NSSI | 33/67 | 66% White, 9% African American, 11% Asian, 6% Multiracial, 5% Other |
| Anestis & Joiner [111] | 492 | 26.99 | 10.33 | 15.45% | Clinical–mental health clinic patients | 41.3/55.1/3.7 transgender | 57% White, 10% African American, 8% Hispanic, 5% Other, 21% did not report |
| Anestis et al. [112] | 358 | 26.91 | 10.13 | 16.20% | Clinical–mental health centre patients | 40.8/59.2/0.3 transgender | 69% White, 11% African American, 2% Native American, 3% Asian, 8% Hispanic, 1% Other |
| Anestis et al. [58] | 176 | 36.12 | 10.33 | 3.41% | Clinical—substance use disorder in patients with BPD | 64.2/35.8 | 54% White, 38% African American, 5% Native American, 3% Other |
| Anestis et al. [106] | 934 | 27.05 | - | 8.24% | U.S. military | 77.7/22.3 | 58% White, 24% African American, 4% Hispanic, 6% Other |
| Aschrafi et al. [142] | 13 | 40.00 (control) | 8.70 | 38.46% | Male suicide by hanging and diagnosis of MDD | 100/0 | Hungarian sample |
| | | 51.80 (experimental) | 6.50 | | | 100/0 | |
| Baer et al. [99] | 365 | 38.79 | 11.6 | 14.80% | Community—substance users recruited from Amazon's Mechanical Turk | 40.8/59.2/0.3 transgender | 82% White, 10% African American, 2% Native American, 6% Asian, 2% Hispanic, 3% Other |
| Baertschi et al. [104] | 167 | 33.60 | 14.6 | 63.47% | Presented at emergency department for a suicide-related event | 39.3/61.7 | Not reported |
| Ben-Efraim et al. [121] | 1,276 | - | - | 100% | Suicide attempters and their parents | 45.5/54.5 | 100% White |
| Beyond Blue [44] | 35 | 43.00 | - | 100% | Australian males | 100/0 | 94% White, 6% First Nations People |
| Biddle et al. [45] | 22 | 36.02 | 11.45 | 100% | Attempted suicide within previous two years | 54.5/45.5 | Not reported |
| Blankenship [64] | 426 | 36.02 | 11.45 | 7.98% | Community sample with and without trauma history | 43.7/56.1/0.2 transgender | 79% White, 6% African American, 1% Native American, 5% Asian, 6% Multiracial, 3% Hispanic, 1% Other |
| Brackman et al. [120] | 113 | 19.00 | 4.33 | 6.19% | University students | 31.9/68.1 | 75% White, 12% Hispanic, 13% Other |
| Calati et al. [76] | 400 | 39.20 | 13.60 | 27.75% | Clinical—Affective spectrum, schizophrenia spectrum, BPD, MDD, BP | 38.7/61.3 | German sample |
| Calati et al. [76] | 70 | 42.90 | 14.40 | 25.71% | | 44.3/55.7 | Italian sample |
| Cao et al. [84] | 484 | 60+ | - | 50.00% | Community | 55.8/44.2 | Not reported |
| Carli et al. [130] | 1,265 | 39.61 | 10.53 | 12.89% | Incarcerated males | 100/0 | Not reported |
| Cheek et al. [72] | 10,203 | 25.46 | - | 12.32% | Substance use and MDD histories | 44.0/56.0 | 82% White, 5% African American, 1% Native American, 2% Asian, 9% Hispanic |

*(Continued)*

**Table 4.** (*Continued*)

| Reference | Sample size | Mean age (years) | Standard deviation (years) | Percentage of sample attempters/ suicides | Population | Male/Female/ Other % | Ethnicities[1] |
|---|---|---|---|---|---|---|---|
| Chelmardi et al. [67] | 909 | 22.40 | 3.80 | - | University students | 30./69.7 | Iranian sample |
| Chu et al. [62] | 3,377 | 29.92 | 4.94 | - | U.S. military | 91.9/8.1 | 65% White, 15% African American, 1% Native American, 3%Asian, 13% Hispanic, 2% Native Hawaiian |
| Chu et al. [77] | 973 | 29.94 | 11.33 | 23.74% | U.S. military | 78.8/21.2 | 64% White, 20% African American, 1% Native American, 3% Asian, 13% Multiracial |
| Copeland et al. [89] | 1,420 | Annually aged 9–16, then at ages 19, 21, 24 to 26, and 30. | - | 4.86% | Community sample from predominately rural counties in North Carolina, U.S. | 51.0/49.0 | 73% White, 7% African American, 25% Native American |
| Daruwala et al. [105] | 953 | 27.06 | 8.11 | 3.25% | U.S. military | 82.3/17.7 | 62% White, 27% African American, 11% Other |
| DeVille et al. [126] | 102 | 33.00 (control) | 10.00 | 33.33% | Clinical—MDD, anxiety disorders, PTSD, substance use disorder, alcohol use disorder, eating disorder | 30.0/70.0 | Not reported |
| | | 31.00 | 11.00 | | | 44.0/56.0 | |
| Dhingra et al. [68] | 665 | 24.20 | 8.11 | 24.21% | University students | 28.5/71.5 | 79% White, 3% Black, 12% Asian, 3% multiracial, 2% Other |
| Duddin & Raynes [46] | 75 | - | - | 100% | Death by suicide on the railway | 69.3/31.7 | 89% White, 11% Asian |
| Feltrin et al. [93] | 98 | 25.80 | - | 100% | University hospital residents | 12.0/88.0 | Not reported |
| Forrest et al. (Study 1) [56] | 106 | 30.65 | 11.31 | 28.30% | Clinical—Seeking therapy or assessment services from a psychology clinic | 50.0/50.0 | 72% White, 6% African American, 2% Native American, 2% Asian, 5% Other |
| Forrest et al. (Study 2) [56] | 595 | 26.61 (control) | 10.34 | 22.86% | | 40.7/59.3 | 70% White, 13% African American, 15%Asian |
| | | 30.04 (experimental) | 11.77 | | | 23.5/76.5 | |
| Govind [47] | 24 | 29.33 | - | 100% | Clinical–psychiatric clinic and treatment care centre | 29.17/70.73 | South African sample |
| Hardt et al. [82] | 1,000 | 39.00 | 15.25 | 17.30% | Community | 44.3/55.7 | Polish and German sample |
| Heiden-Rootes et al. [97] | 23,232 | - | - | 5.95% | Community—transgender and nonbinary adults | Cross-dresser: 7.3% Transgender woman: 51.7% Transgender man: 22.0% Nonbinary/ genderqueer: (birth-assigned female) 12.9% Nonbinary/ genderqueer: (birth-assigned male) 6.0% | 71% White, 14% African-American, 15% Hispanic |
| Hsiao et al. [139] | 1 | 63 | - | 100% | Clinical male sample– depression | 100/0 | Not reported |

(*Continued*)

**Table 4.** (Continued)

| Reference | Sample size | Mean age (years) | Standard deviation (years) | Percentage of sample attempters/ suicides | Population | Male/Female/ Other % | Ethnicities[1] |
|---|---|---|---|---|---|---|---|
| Huang et al. [117] | 954 | 26.30 | 7.11 | 66.56% | International community sample | 27.3/67.7/4.7 did not answer | 80% White, 4% African American, 6% Asian, 5% Hispanic, 1% Native American and Indigenous People, 5% Other |
| Joiner et al. (Study 2) [115] | 313 | 22.17 | 2.76 | 39.94% | Clinical—MDD, BP, anxiety disorder, schizophrenia spectrum | 82.1/17.9 | 60% White, 25% African American, 2% Native American, 1% Asian, 10% Hispanic |
| Jordan et al. [98] | 245 | 29.06 | 6.35 | 100% | Clinical—schizophrenia spectrum, MDD, BP, alcohol disorder, substance use disorder | 53.5/46.5 | 77% White, 23% Non-White |
| Jordan & Samuelson [101] | 690 | 24.10 | 12.61 | 49.42% | Community | 33.3/66.7 | 72% White, 10% African American, 3% Asian, 15% Hispanic |
| Jovičić et al. [127] | 251 | 49.13 | 13.09 | 33.07% | Clinical–recurrent depressive disorder or MDD, single episode | 43.4/56.6 | Serbian sample |
| Kasen et al. [128] | 770 | 13.7, 16.1, and 22.0 | 2.6, 2.8, and 2.7 | 8.83% | Community | 51.0/49.0 | 91% white, 8% African American |
| Kene [78] | 100 | 35.84 | 11.44 | 60% | Clinical—schizophrenia spectrum, BP, MDD, psychotic disorder, substance use disorder, antisocial, narcissistic, BPD | 63/37 | Control group: 53% White, 43% African American, 5% Other. Experimental group: 72% White, 20 African American, 7% Hispanic, 2% Other |
| Kerbrat et al. [86] | 733 | 25.07 | 5.83 | 53.07% | U.S. military | 66.1/33.9 | 57% White, 12% African American, 1% Native American, 4% Asian, 9% Multiracial, 18% Hispanic |
| Khazem & Anestis [17] | 378 | 36.09 | 10.69 | 35.54% | Community | 45.2/54.8 | 75% White, 8% African American, 10% Asian, 5% Hispanic |
| Kishikawa et al. [140] | 2 | Japan: 62 years U.S.: 84 years | - | 100% | Clinical–males with Alzheimer's disease | 100/0 | Japan and U.S. participants |
| Klonsky et al. (Study 3) [116] | 1,656 | 20.07 | 2.00 | 7.00% | University students | 44.0/56.0 | 43% White, 7% African American, 35% Asian, 9% Hispanic, 7% Other |
| Klonsky et al. (Study 4) [116] | 439 | 55.50 | 16.60 | 2.96% | | 39.0/61.0 | 86% White, 6% African American, 1% Native American, 3% Asian, 1% Hispanic |
| Knowles et al. [124] | 1,897 | 42.14 (control) | 13.06 | 8.28% | Community | 40.6/59.4 | Mexican-American sample |
| | | 40.99 (experimental) | 15.93 | | | 29.3/70.7 | |
| Koweszko et al. [144] | 48 | 35.70 | 11.40 | 33.33% | Clinical—substance use disorder, schizophrenia spectrum, depressive disorders, anxiety disorders, personality disorders | 52.1/47.9 | |
| Kunde et al. [48] | 18 | 53.00 | 13.40 | 100% | Male farmers | 100/0 | Australian |

(*Continued*)

**Table 4.** (Continued)

| Reference | Sample size | Mean age (years) | Standard deviation (years) | Percentage of sample attempters/ suicides | Population | Male/Female/ Other % | Ethnicities[1] |
|---|---|---|---|---|---|---|---|
| Law & Anestis (Study 1) [103] | 124 | 20.86 | 8.87 | 4.84% | University students | 17.2/82.8 | 66% White, 28% African American, 3% Hispanic, 3% Other |
| Law & Anestis (Study 2) [103] | 84 | 20.87 | 5.51 | 7.14% | | 21.4/78.6 | 63% White, 28% African American, 1% Native American, 4% Hispanic, 5% Other |
| Law et al. (Study 2) [118] | 99 | 23.63 | 8.16 | 35.35% | University students | 23.2/76.8 | 40% White, 44% African American, 10% Asian |
| Leira et al. [71] | 64,770 | 50.20 | - | 0.14% | Community | 46.9/53.1 | Norwegian sample |
| Li et al. (2017) [122] | 162 | 37.60 | 14.50 | 45.68% | Clinical—schizophrenia spectrum | 66.0/34.0 | 100% White |
| Liu [74] | 4,882 | - | - | 0.92% | Community youth | 49.4/50.6 | 69% White, 5% Asian |
| Love & Durtschi [73] | 4,208 | Between 25–34 | - | 8.10% | Community | 44.6/55.4 | 63% White, 24% African American, 4$ Native American, 4% Asian, 4% Hispanic |
| Martin [52] | 1 | 61; actual age | - | 100% | Male—Hemingway, E. M. | 100/0 | White |
| McCarthy et al. [138] | 138 | - | - | 32.61% | Incarcerated males | 100/0 | 62% White, 30% African American, 8% Native American |
| Medeiros et al. [134] | 62 | 41.27 (control) | 11.28 | 51.61% | Clinical—MDD Clinical—eating disorder | 13.3/86.7 | Portuguese sample |
| | | 38.16 (experimental) | 9.47 | | | 25.0/75.0 | |
| Miller [114] | 50 | 35.98 | 11.52 | 100% | Community | 24.0/72.0/4.0 transgender | 76% White, 2% African American, 8% Asian, 8% Multiracial, 6% Hispanic |
| Oakes-Rogers & Slade [92] | 32 | 30.20 | 7.60 | 100% | Incarcerated females | 0/100 | U.K. sample |
| Olié et al. [137] | 343 | - | - | 49.56% | Clinical–BP | 43.1/59.9 | French sample |
| Oshnokhah et al. [143] | 90 | 26.70 | 0.70 | 55.56% | Clinical—hospitalised suicide attempters | - | Kurdish sample |
| Pelton et al. [113] | 695 | 41.60 | 12.27 | 23.89% | Clinical—autism diagnosis | 35.1/61.7/3.2 other gender | Not reported |
| Pettit et al. [88] | 298 | 22.22 | 2.76 | 41.28% | U.S. Military | 82.2/17.8 | 63% White, 24% African American, 8% Hispanic |
| Pisetsky et al. [81] | 114 | 33.70 | 12.11 | 21.05% | Clinical—eating disorder | 6.1/93.9 | Not reported |
| Pitman et al. [131] | 14,648 | 44.00; median age (control) | - | 100% | Clinical—schizophrenia spectrum, affective spectrum, substance use disorder, alcohol use disorder, personality | 66.9/33.1 | 93% White, 8% African American |
| | | 53.00; median age (experimental) | - | | | 64.2/35.8 | |
| Price [80] | 490 | 37.00 (control) | - | 71.43% | Incarcerated males | 100/0 | 36% White, 36% African American, 9% Multiracial, 19% Hispanic |
| | | 40.00 (experimental) | - | | | 100/0 | 44% White, 11% African American, 6% Asian, 7% Multiracial, 26% Hispanic |
| Rappaport et al. [75] | 11,647 | 47.68 (control) | 5.59 | 3.63% | Clinical females—MDD (no BP, psychosis, intellectual disability, or alcohol or substance use prior to first major depressive episode) | 0/100 | Chinese sample |
| | | 44.44 (experimental) | 8.94 | | | 0/100 | |

(*Continued*)

**Table 4.** (Continued)

| Reference | Sample size | Mean age (years) | Standard deviation (years) | Percentage of sample attempters/ suicides | Population | Male/Female/ Other % | Ethnicities[1] |
|---|---|---|---|---|---|---|---|
| Raubenheimer & Jenkins (Focus group) [49] | 5 | - | - | 100% | Clinical–females hospitalised from suicide attempt | 0/100 | South African sample |
| Richard-Devantoy et al. [136] | 60 | 76.07 | 5.77 | 33.33% | Clinical—current major depressive episode | 38.3/61.7 | Not reported |
| Rogers et al. [57] | 540 | 35.94 | 11.41 | 7.96% | Community | 44.2/55.6/0.2 transgender | 78% White, 9% African American, 1% Native American, 9% Asian, 7% Hispanic, < 1% Pacific Islander, Other |
| Ryan et al. [83] | 245 | 22.80 | 1.40 | 30.20% | Community–self-identified as lesbian, gay, bisexual, and/or transgender | 46.5/44.9/8.6 transgender | 49% White, 51% Hispanic |
| Shelef et al. [94] | 167 | 19.70 | 1.00 | 34.73% | Israeli defence force personnel | 59.5/40.5 | Israeli sample |
| Shelef et al. [87] | 60 | - | - | 100% | Israeli defence force personnel | 60.0/40.0 | Israeli sample |
| Shim et al. [109] | 200 | 74.90 | - | - | Individuals aged over 65 years at welfare centers for older persons | 41.5/58.5 | South Korea sample |
| Smith et al. [90] | 44 | 33.07 (ideators) | 14.03 | 34.09% | University students | 33.3./66.7 | 80% White, 7% Native American, 7% Asian, 7% Hispanic, 7% Other |
| | | 28.6 (attempters) | 11.98 | | | 46.7/53.3 | 67% White, 13% Native American, 7% Asian, 7% Hispanic, 7% Other |
| Smith et al. (Study 1) [63] | 204 | 25.67 | 8.85 | 27.60% | Females recruited through eating disorder clinics and the community | 0/100 | 91% White, 3% African American, 3% Asian, 2% Hispanic, 2% Other |
| Smith et al. (Study 4) [63] | 512 | 18.89 | 2.70 | - | University students | 20/80 | 79% White, 14% African American, 1% Native American, 5% Asian, 13% Hispanic |
| Smith et al. [102] | 30 | 30.83 | 13.02 | 50.00% | Community sample of depressed individuals | 40.0/60.0 | 73% White, 7% Native American, 7% Asian, 7% Hispanic, 7% Other |
| Sokolowski et al. [79] | 1,179 | 35.70 (Males—control) | 16.10 | 55.98% | Suicide attempters and both their parents | 45.9/54.1 | Ukrainian and Russian sample |
| | | 34.60 (Females—control) | 14.8 | | | | |
| | | 24.60 (Males–experimental) | 7.30 | | | 51.1/48.9 | |
| | | 23.80 (Females—experimental) | 7.10 | | | | |
| Stoliker & Abderhalden [100] | 548 | 35.64 | 10.81 | 20.80% | Incarcerated individuals | 72.6/27.4 | Only 41% White reported |
| Sunnqvist et al. [51] | 23 | 41; median age | - | 100% | Clinical—BP, MDD, dysthymia, depression disorder not specified, substance use disorder, adjustment disorder, anxiety disorder | 65.2/34.8 | Swedish sample |
| Suto & Arnaut [50] | 24 | 31.83 | 1.01 | 100% | Incarcerated individuals | 87.5/12.5 | 71% White, 21% Multiracial, 8% Hispanic |

(*Continued*)

**Table 4.** (Continued)

| Reference | Sample size | Mean age (years) | Standard deviation (years) | Percentage of sample attempters/ suicides | Population | Male/Female/ Other % | Ethnicities[1] |
|---|---|---|---|---|---|---|---|
| Tull et al. [123] | 59 | 31.59 | 10.09 | 28.81% | Clinical–BPD | 52.5/47.5 | 86% White, 14% African American |
| Van Orden et al. (Study 2) [60] | 228 | 26.21 | 9.56 | 15.36% | Clinical—mood disorder, anxiety disorder, substance use disorder, personality disorder, schizophrenia spectrum, other (e.g., eating disorder, trichotillomania) | 44.9/55.1 | 74% White, 12% African American, 9% Hispanic, 5% Other |
| Van Orden et al. (2008) [60] | 153 | - | - | 14.38% | | - | Not reported—Similar to above |
| Van Orden et al. [91] | 172 | 68.02 | 13.20 | 50.00% | Death by suicide | 73.3/26.7 | 98% White, 2% Other |
| Wolford-Clevenger et al. [108] | 396 | 34.55 | 10.73 | 25.00% | Community sample of individuals arrested for domestic violence and mandated to Batterer Intervention Programs | 78.8/21.2 | 67% white, 10% African American, 2% Native American, 1% Asian, 13% Hispanic, 8% Other |
| Wolford-Clevenger et al. [65] | 134 | 32.50 | 8.21 | 28.36% | Females seeking shelter from interpersonal violence | 0/100 | 41% White, 53% African American, 3% Native American, 3% Other |
| Wong et al. [129] | 170 | - | - | 50.00% | Death by suicide aged 30–49 years | 62.4/37.6 | Hong Kong sample |
| Yang et al. [69] | 1,097 | 19.77 | 1.67 | 3.83% | University students | 43.8/56.2 | Chinese sample |
| Zhao et al. [95] | 617 | 32.80 | 13.40 | 100% | Clinical | 25.9/74.1 | Chinese sample |

[1] Values may not add up to 100% because of rounding

*Note.* Anestis et al. [133] and Law et al. [132] not included because they are ecological studies that include total country or city populations.

A dash indicates data not reported.

Abbreviations: Non-Suicidal Self-Injury (NSSI); Borderline Personality Disorder (BPD); Major Depressive Disorder (MDD); Bipolar Disorder (BP); Post-Traumatic Stress Disorder (PTSD).

nine qualitative studies comprised suicide attempters or deaths by suicide (e.g., suicide notes). Almost half of the research had been published since 2017 ($n = 43$) and this publication date trend was similar across and within contributors. Table 5 displays groupings of studies based on contributors of suicide capability [9, 16].

**Acquired contributors.** Overall, 55 out of the 90 included studies were allocated to the acquired capability contributor grouping. Most work has been done looking at painful and provocative events ($n = 31$), with fearlessness about death being the least researched ($n = 4$). The majority of studies associated with the three sub-factors were conducted over 5 years ago ($n = 30$) with painful and provocative event studies being the most recent studies published ($n = 14$).

*Painful and provocative events*. Most of the painful and provocative event studies identified in this review sought to identify relationships between painful and provocative events and suicide attempts ($n = 11$), or relationships between painful and provocative events and acquired capability ($n = 14$). Events ranged from emotional abuse [49] to general painful and provocative events [60], with most studies either using community (e.g., [82–84]), clinical (e.g., [51, 60, 85]), or military samples (e.g., [86–88]).

There were seven comparison studies, and all found that painful or provocative events differentiated suicide attempters from suicide ideators and/or controls, with most ($n = 5$) being published before 2017. Care-giver perpetrated abuse [73], childhood maltreatment [82, 89],

**Table 5. Study sub-factor allocation.**

| Capability Contributor | Sub-factor | Studies allocated to sub-factor |
|---|---|---|
| Acquired (*n* = 55) | Combination of painful and provocative events and fearlessness about death (*n* = 20) | Allbaugh et al. [135]; Ammerman et al. [119]; Anestis & Joiner [111]; Anestis et al. [112]; Brackman et al. [120]; Chu et al. [62]; Chu et al. [77]; Huang et al. [117]; Joiner et al. [115]; Kene [78]; Khazem & Anestis [17]; Klonsky et al. [116]; Law et al. [118]; Miller [114]; Pelton et al. [113]; Pisetsky et al. [81]; Price [80]; Shim et al. [109]; Wolford-Clevenger et al. [108]; Zhao et al. [95]. |
| | Painful and provocative events (*n* = 31) | Anestis et al. [58]; Baer et al. [99]; Beyond Blue [44]; Blankenship [64]; Cao et al. [84]; Cheek et al. [72]; Copeland et al. [89]; Feltrin et al. [93]; Govind [47]; Hardt et al. [82]; Heiden-Rootes et al. [97]; Jordan & Samuelson [101]; Jordan et al. [98]; Kerbrat et al. [86]; Law & Anestis [103]; Love & Durtschi [73]; Oakes-Rogers & Slade [92]; Pettit et al. [88]; Raubenheimer & Jenkins [49]; Rogers et al. [57]; Ryan et al. [83]; Shelef et al. [94]; Shelef et al. [87]; Smith et al. [90]; Smith et al. [63]; Smith et al. [102]; Stoliker & Abderhalden [100]; Sunnqvist et al. [51]; Van Orden et al. [60]; Van Orden et al. [91] Wolford-Clevenger et al. [65]. |
| | Fearlessness about death (*n* = 4) | Anestis et al. [106]; Baertschi et al. [104]; Daruwala et al. [105]; Suto & Arnaut [50]. |
| Dispositional (*n* = 13) | Genetic influences (*n* = 6) | Ben-Efraim et al. [121]; Calati et al. [76]; Knowles et al. [124]; Li et al. [122]; Sokolowski et al. [79]; Tull et al. [123]. |
| | Interoceptive deficits (*n* = 2) | DeVille et al. [126]; Forrest et al. [56]. |
| | Personality traits (*n* = 5) | Carli et al. [130]; Jovičić et al. [127]; Kasen et al. [128]; Rappaport et al. [75]; Wong et al. [129]. |
| Practical (*n* = 8) | Access to lethal means (*n* = 2) | Anestis et al. [133]; Law et al. [132]. |
| | Knowledge of lethal means (*n* = 1) | Liu [74]. |
| | Combination of access and knowledge (*n* = 5) | Biddle et al. [45]; Duddin & Raynes [46]; Kunde et al. [48]; Leira et al. [71]; Pitman et al. [131]. |
| Cognitive (*n* = 10) | Executive functioning deficit (*n* = 7) | Abdollahpour Ranjbar et al. [135]; Hsiao et al. [139]; Kishikawa et al. [140]; McCarthy et al. [138]; Medeiros et al. [134]; Olié et al. [137]; Richard-Devantoy et al. [136]. |
| | Neurological responses to stress (*n* = 3) | Aschrafi et al. [142]; Koweszko et al. [144]; Oshnokhah et al. [143]. |
| Suicide capability (*n* = 4) | Acquired/dispositional/practical | Chelmardi et al. [67]; Dhingra et al. [68]; Martin [52]; Yang et al. [69]. |

intravenous substance use [72], psychological distress [58], and frequent experiences of painful and provocative events [90, 91] were found to differentiate suicide attempters from suicide ideators and/or controls. However, these studies used events as indicators of suicide capability and did not include a measure of acquired capability. Therefore, the relationship between acquired capability and these painful and provocative events is somewhat unclear.

Of the 11 studies that sought to identify potential relationships between painful and provocative events and suicide attempts most reported significant relationships (*n* = 10) with most studies again being published prior to 2017 (*n* = 6). Emotional abuse [49] was reported to contribute to suicide attempts, with trauma involving violence being related to multiple suicide attempts before death by suicide [92]. Six of the seven studies that looked at accumulation of life stressors and suicide attempts reported significant relationships [44, 84, 87, 93–95]. However, Pettit et al. [88] did not find a relationship between life stressors and suicide attempts within a sample of military personnel diagnosed with early onset BP. Sexual orientation and gender identity change efforts, which have previously been reported as a painful injury [96], were found to be related to both suicidal ideation and suicide attempts [83, 97]. These results suggest that life stressors, abuse, and trauma appear to contribute to suicide capability. Like the comparison studies, without a measure of acquired capability it is difficult to come to any clear conclusion concerning the relationship between painful and provocative events and the acquired contributor.

Of the 14 studies that investigated potential relationships between painful and provocative events and acquired capability, more than half (*n* = 8) were published in the past five years

with 11 studies reporting significant relationships. Events include generally experiencing more painful and provocative events [60], traumatic experiences including NSSI [98], continuing to exercise despite pain and exhaustion [57, 63], substance use [99] and childhood abuse [85]. Studies looking at perpetrators of violence found perpetrators had an increased acquired capability [100] compared to victims of violence [94, 101]. Acquired capability was found to increase with combat deployments [86] and the use of maladaptive coping strategies [51]. However, Smith et al. [102] did not find that the rate of physiological habituation contributed to the development of acquired capability. Similarly persisting through pain did not impact acquired capability [103], and trauma involving violence was found to be unrelated to acquired capability [64]. Despite these non-significant results, overall results suggest that painful and provocative events appear to contribute to an acquired capability of suicide.

*Fearlessness about death*. Only four studies were identified that operationalised acquired capability as fearlessness about death and half of these studies had been published in the previous five years [104, 105]. One study that compared suicide attempters and suicide ideators who presented to an emergency department found that fearlessness about death did not differentiate between the two groups [104]. One relationship study found that sensation seeking was related to fearlessness about death [105] and another found that fearlessness about death only correlated with lifetime suicide attempts at high levels of fearlessness about death, but not at mean or low levels [106]. A qualitative study found that Buddhist beliefs can help overcome fearlessness about death [50], arguing that this is based on the Buddhist belief that suicide can be a noble act [107]. The three quantitative relationship studies indicate that there appears to be ambiguity about the relationship between aspects of fearlessness about death and suicide capability at this point in time. However, the paucity of research makes it premature to draw any firm conclusions about this capability contributor.

The seven studies that sought to compare suicide attempters to suicide ideators and/or controls found that painful and provocative events were able to distinguish suicide attempters from ideators and/or controls. However, this did not carry over to the one fearlessness about death study that sought to compare suicide attempters and suicide ideators. Eleven of the 14 studies that aimed to find potential relationships between acquired capability and painful and provocative events did find such a relationship. Suicide attempts and death by suicide were also found to be associated with painful and provocative events such as abuse, trauma, and life stressors, and high (not average or low) levels of fearlessness about death. These results suggest that painful and provocative events may potentially influence acquired capability more than fearlessness about death.

*Acquired contributor as a combination of painful and provocative events and fearlessness about death*. The trend of more studies focusing on potential relationships between the acquired contributor and suicide attempts ($n = 16$) rather than seeking to differentiate between suicide ideators and attempters ($n = 4$), continued when the acquired contributor was considered as a combination of painful and provocative events and fearlessness about death. Of these 20 studies, over half ($n = 11$) were published more than 5 years ago with mixed findings being reported across studies.

The ACSS [60] is a measure of acquired capability that combines both painful and provocative events and fearlessness about death and has been used in four differentiation studies. The combined acquired contributor did not differentiate between suicide attempters and suicide ideators in two studies [78, 108]. However, studies that looked at each sub-factor contained within the ACSS found that painful and provocative events differentiated suicide attempters from suicide ideators whereas fearlessness about death did not [17, 81]. This provides further support for the proposition that painful and provocative events appear to distinguish suicide attempers from suicide ideators more so than fearlessness about death.

Four studies have used variations of the ACSS to examine the relationship between the combination of acquired contributors and suicidality with varying results. Using an adapted version of the ACSS, support was found for the combined acquired contributor as a moderator of the movement from suicide ideation to suicide attempt [109]. However, shortened versions of the ACSS found that negative urgency (i.e., acting rashly to reduce negative feelings [110]) was [111] and was not [112] reported in the development of acquired capability. Further complicating our understanding is Pelton et al. [113] who reported a significant relationship to suicide attempt history only when painful and provocative events (i.e., traumatic life events) were combined with the ACSS-FAD [61]. Similar to previous findings, fearlessness about death appears to be less related to the acquired contributor than painful and provocative events.

According to the IPTS [10, 11], an individual requires an increased acquired capability that is characterised by fearlessness about death and increased pain tolerance to attempt suicide. Therefore, previous suicide attempts are considered to contribute to the acquired contributor as a combination of the two sub-factors. Multiple suicide attempts were reported by suicide attempt survivors [47, 114] and were found to be related to acquired capability [80, 115]. These results suggest attempting suicide may contribute as a combined acquired contributor.

Despite NSSI being suggested as a critical risk factor for suicide, results from seven studies mostly published before 2017 ($n = 4$) indicate that there is a lack of consistent evidence for how NSSI may contribute to suicide capability. For example, historical NSSI [62, 77] was found to be related to acquired capability in both suicide ideators and suicide attempters [116, 117]. Yet NSSI frequency (i.e., frequency, severity, and duration of NSSI) was found to be related to individuals with suicide attempt history [118] more so than individuals who had not attempted suicide [119]. However, it was not related to the acquired constructs of pain tolerance and fearlessness about death [120]. Taken together, findings from the acquired capability as a combination of the two constructs are similar to reported results when painful and provocative events and fearlessness about death were researched separately. This further suggests it may be that painful and provocative events rather than fearlessness about death that are more related to suicide attempts and the acquired contributor.

**Dispositional contributors.** Thirteen studies were allocated to the dispositional contributor grouping. The focus of research was mostly on potential genetic associations ($n = 6$), or personality traits ($n = 5$), with six studies being published since 2017.

*Genetic associations.* Five of the six genetic studies focused on polymorphisms with most studies finding polymorphisms that potentially contribute to the dispositional contributor. Four studies compared suicide attempers to suicide ideators and/or controls. Both studies published before 2017 distinguished suicide attempters from controls by combining single nucleotide polymorphisms rs16940665 [121] and rs7559979 [79] with painful and provocative events. Two more recent studies looked at polymorphisms without combining other factors. Males (but not females) were more likely to have attempted suicide compared to controls if they carried the single nucleotide polymorphism rs300774 [122]. Similarly, Val/Val carriers of the catechol-o-methyltransferase Val158Met polymorphism were also more likely to report a previous suicide attempt compared to suicide ideators and controls [123].

Results from one of two relationship studies reported relationships with suicide attempters. Reduced brain cholesterol metabolism was found in suicide attempters [124], but no associations were found between seven catechol-o-methyltransferase single nucleotide polymorphisms and suicide attempters with various psychiatric disorders [76]. Given the scant research that has been conducted in this area, it is not possible at this point in time to conclude whether genetics contributes to suicide capability.

*Interoception.* Interoception refers to the process of how the brain receives signals from the body enabling us to feel physical states (e.g., pain, hunger, etc.) and emotional states (e.g., fear,

calmness, etc.) [125]. Two studies were conducted and found that interoceptive deficits differentiated suicide attempters from suicide ideators [56, 126]. Different measures were used in these studies making comparisons across studies problematic. In addition, the small number of studies limits what can be said about interoception with any confidence.

*Personality traits*. Research on personality traits identified in this review focused on neuroticism (*n* = 2) or impulsivity (*n* = 3), with comparison study designs reporting different findings to association studies. Both studies that focused on neuroticism were published in the past five years. One comparison study found that suicide attempters scored higher on neuroticism than controls [127]. A different study focusing on a sample of females with a MDD diagnosis found neuroticism was related to suicidal ideators but not suicide attempters [75]. The most recent study on impulsivity identified in this review was published 10 years ago. Two comparison studies found that impulsivity was higher in suicide attempters [128] and individuals who died by suicide [129] when compared to controls. However, Carli et al. [130] found that impulsivity did not contribute to suicide attempts among male incarcerated individuals. As with other dispositional capability contribution studies, little can be concluded about the role of neuroticism and impulsivity at this point in time. Overall, the limited number of studies makes it difficult to come to any firm conclusions about dispositional contributors of suicide capability.

**Practical contributors.** This is the contributor where the least number of studies have been conducted (*n* = 8). Most studies were association studies (n = 5) and all were published in the past five years (*n* = 5). Three of the five studies that explored knowledge of and access to lethal means were qualitative. Perceptions of an expected certain and quick death, and accessibility were prominent themes when choosing hanging as a method to attempt suicide [45] and the railway to die by suicide [46]. Farmers who died by suicide had experience with and ready access to firearms, which helped towards choosing firearms as the method [48]. The following two association studies also explored knowledge of and access to lethal means. One study reported that prescribed medications increased suicide deaths by overdose among individuals with psychiatric diagnoses and co-morbid physical illness compared to controls with psychiatric diagnoses but without co-morbid physical illness [131]. Similarly, a critical difference between fatal and non-fatal suicide attempts was whether the method was violent (i.e., hanging) or not (i.e., overdose) [71], however it is unknown how the choice of method was determined.

Knowledge of or access to lethal means were individual focuses of three association studies. One study looked at knowledge of lethal means and found that exposure to suicide was associated with both suicide ideation and suicide attempts [74]. Restricting accessibility by placing barriers on a bridge [132] and firearm background checks or mandatory waiting period legislations [133] appeared to reduce deaths by suicide on nearby city bridges and at a U.S. state level, respectively. However, it is difficult to know if other prevention strategies were implemented that may have contributed to the reduction in suicides. Given this contributor has the least number of studies published, it is not possible to draw strong conclusions about the role of practical contributors towards suicide capability.

**Cognitive contributors.** The cognitive contributors cluster is a new contributory cluster identified through this review. This new cognitive contributor is being proposed based on finding 10 studies that focused on cognitive aspects found in suicide attempters but not suicide ideators or controls, and individuals who died by suicide. Most of the research focused on cognitive deficits (*n* = 7) and was published prior to 2017 (*n* = 6). Four studies found suicide attempters had significantly impaired cognitive functioning when compared to suicide ideators [134, 135] and controls [136, 137]. Three association studies found cognitive deficits were reported by suicide attempters [138], including attempters with dementia [139], and identified in individuals who had an Alzheimer's diagnosis and died by suicide [140].

An additional three studies looked at neurological responses associated with stress, two of which were comparison studies. Evidence of cellular differences within a midbrain section (i.e., the Edinger-Westphal nucleus) that has been suggested to regulate neuronal response to stress [141] were found in individuals who died by suicide but not in controls [142]. Further, oxidative stress levels distinguished suicide attempters from controls [143] and oxidative stress components (i.e., higher advanced glycation end products and dityrozine, and negative catalase) were found to be associated with suicide attempt history [144]. Whilst there has been only a small number of studies conducted thus far, similarity of findings suggests that cognitive impairments and neurological responses to stress may potentially contribute to suicide capability.

**Suicide capability as a combination of acquired, dispositional, and practical contributors.** Up to this point, most studies reviewed have examined contributors in isolation to each other, noting the exception of the combined acquired contributor studies. Four studies have looked at suicide capability as a broader multifaceted concept with three being published in the past three years making this is a relatively recent area of investigation.

Two of these four studies sought to compare suicide attempters and suicide ideators. These two studies measured suicide capability as a combination of all three contributors and found that "suicide capacity robustly distinguished" [68 p483, 69 p657] suicide attempters from suicide ideators. Importantly, Yang et al. [69] reported that acquired and dispositional contributors did not independently differentiate between the two groups when measured in isolation. Dhingra et al. [68] reported similar results after controlling for suicidal desire; dispositional contributors considered in isolation did not predict history of suicide attempt, but acquired and practical contributors independently predicted suicide attempt history.

However, results from an association study that sought to combine all factors from the three ideation-to-action theories within a sample of Iranian students are contradictory. The results include support for acquired capability as a predictor of suicide attempt, but also find no support for acquired, or dispositional, or practical capabilities as individual predictors of suicide attempts. For example, the authors state that "acquired capacity . . . had a significant effect on suicide attempt" [67 p10], yet "none of the dispositional, acquired, and practical capacities . . . had any significant effect on suicide attempt" [67 p10]. From this it is unclear if the acquired contributor did or did not predict suicide attempts. Further, it is unknown if any of the contributors had an indirect effect on suicide capability because this was not investigated within the study. Therefore, findings from this study are difficult to make sense of because the same scale (i.e., SCS-3) was used to attain these contradicting results.

The final study was a qualitative study that identified the individual as having an extensive family history of suicide, experience of several injuries, and as having had knowledge of, and access to, lethal means [52]. Given that few studies combine the three contributors of suicide capability as suggested by Klonsky and May [9] and Klonsky et al. [16], there is promising but limited evidence to draw conclusions about suicide capability as a combination of acquired, dispositional, and practical contributors.

## Discussion

The movement from ideation-to-action is largely accepted as being complex and multifaceted [5–9, 15–17], yet most of the research identified in this scoping review is not multifaceted nor does it reflect this complexity. Only four studies looked at more than one contributor despite various calls [7, 18, 19, 21] for research to advance beyond single factor studies. Given the IPTS [10, 11] was the first theory within the ideation-to-action framework, this review unsurprisingly found that the majority of suicide capability research focused on the single

contributor of acquired capability, which is a core component of that theory. Based on the studies reviewed, there appears to be support for a range of painful and provocative events (e.g., childhood abuse, traumatic experiences, and cumulative life stressors) contributing to acquired capability and as a differentiating contributor between suicide attempters and suicide ideators and/or controls. Painful and provocative events appear to have a stronger association with acquired capability than fearlessness about death suggesting that there may be differences in how these sub-factors contribute to the development of acquired capability. For example, fearlessness about death may have an indirect effect on acquired capability whereas painful and provocative events may have a direct effect. Similar to previous findings [7], suicide attempts and NSSI appear to be most indicative of future suicide attempts across different study types and populations. Studies that did not find support for acquired contributors used partial measures [64, 102, 104], or were predominately male samples [78, 108], or less than 10% of the sample included individuals who had previously attempted suicide [103]. Nevertheless, painful and provocative events appear to be related to acquired capability.

In terms of dispositional and practical contributors, the small number of studies identified makes it difficult to ascertain their influence on suicide capability. There are promising results, but more evidence is needed before any firm position on contribution can be made. For example, access to and knowledge of lethal means appears to contribute to fatal and non-fatal suicide attempts, which is similar to May and Victor's [21] review findings. Genetics have been suggested as a potential contributor to suicide capability [19, 20] and the findings from the limited studies in this review indicate that some genetic aspects (e.g., single nucleotide polymorphisms) appear to be related to suicide attempters. However, these findings have not been replicated and thus remain as isolated findings.

An additional potential contributor of suicide capability, cognitive, has been identified in this review. This was an unexpected yet important finding because it builds on previous research [21] that suggests suicide capability is complex and involves more contributors than first conceptualised. These cognitive studies suggest that executive functioning is decreased in suicide attempters and individuals who have died by suicide. Stress has been argued previously to contribute to suicidality [145] and results from this review suggest that both accumulative life stressors and neurological responses to stress may contribute to suicide capability.

When all three contributors of suicide capability were tested in combination as suggested by Klonsky and May [9] and Klonsky et al. [16], and which reflects the premise that suicide is complex and multifaceted, suicide capability was found to differentiate between suicide attempters and suicide ideators [68, 69]. This is important because it was the combination of contributors rather than each individual contributor that differentiated the two groups, which indicates that combining contributors may be better placed to provide greater understanding about the complex movement from ideation-to-action. However, more combination research is needed if we are to fully understand suicide as a multifaceted concept. Suicide is complex [5–9, 15–17] yet most research identified in this review is not. Multi-contributor research more accurately reflects the clinical reality that clients are likely to present with a multitude of factors [146] and that understanding how these factors interplay with each other, rather than trying to identify a single isolated contributor is more likely to lead to good clinical outcomes. Therefore, understanding the interplay of contributors of suicide capability will likely help towards bridging the gap between research and clinical practices with suicidal individuals.

## Limitations of the literature

A key limitation identified from this review is the large number of single contributor studies. Single contributors only offer a portion of understanding towards suicide capability [11] and it

is surprising to see research not utilising multiple contributors. However, the combination of contributors is a relatively recent theoretical development, and it is therefore possible that such research is currently being conducted or may be conducted in the future. An example is the amount of research on individual painful and provocative events that does not combine fearlessness about death. This lack of combination may be adding to the ambiguity about how these two constructs potentially contribute to the development of acquired capability. Without combining these two constructs' questions remain about how these two mechanisms potentially influence each other for the development of acquired capability. Moreover, single contributor studies risks repeating the unproductive conclusion that almost all negative life events are risk factors for suicide [6]. This lack of differentiation amongst risk factors was a key foundation for the need for the ideation-to-action framework [5], and yet this review suggests that the existing research within the ideation-to-action framework is falling prey to the same criticism by continuing the focus on single variables. Not every individual who experiences painful and provocative events or has certain dispositions or access to lethal means will attempt suicide [8]. Rather it is more likely that it is the combination of these contributors that explains the transition from thinking about suicide to attempting suicide. It is this combination research focus that is needed to advance the field. This supports Franklin et al.'s [7] recommendations that to better understand the many complex pathways to suicide attempts, research would benefit from shifting from a single contributor focus to a multiple contributor focus.

A second limitation identified is that only a small number of studies have been conducted on contributors that are not the acquired contributor. Results are promising, however the limited evidence available within each contributor raises questions about how contributors potentially function within the concept of suicide capability. Similar to previous [20, 21] findings, practical contributors may play a role in suicide capability, but given the limited evidence we can only speculate about the impact that knowledge of and access to lethal means has for the movement from ideation-to-action. Adding to this uncertainty is what appears to be a sporadic and disconnected approach to building the evidence base. Study publication dates for each contributor were similar in that close to half of studies were published before 2017, raising questions about whether studies without significant results have been impacted by publication bias. Early non-significant findings that are not able to be published may lead to contributors of potential interest being abandoned in favour of known contributors that have found support and thus are more likely to be published. Most studies identified within this review contained significant findings and whilst this is encouraging, the lack of non-significant publications has potentially influenced the suicide capability field that appears to be absent of strategic development for contributors. This absence is likely to be preventing progress as it is necessary to take strategic steps based on both significant and non-significant evidence to build incremental knowledge and advance theoretical understanding [147]. The apparent piecemeal approach further indicated by isolated research and lack of replication is likely to also be hampering theoretical progress. Replication is necessary to challenge existing knowledge with new evidence to sharpen conceptual contours [148]. This is particularly important for understanding suicide capability given the contours of each contributor may not be as sharp as needed and are still evolving. Current research has resulted in a lot of theoretical research that isn't really telling us anything new and thus not really helping to enhance the practical applicability of this research.

An additional three limitations have been identified by this review. First, the majority of studies reviewed were cross-sectional and therefore only capture specific time snapshot of capability. Consequently, how an individual develops suicide capability remains unknown. Furthermore, only 43% of studies ($n = 39$) were appraised as high quality with potential confounding factors being commonly overlooked or not addressed. Second, the inconsistent use of measures makes it difficult for results to be compared and contrasted across studies, thus

limiting not only the applicability of the results but the ability of these studies to provide meaningful contributions to the field. Although there is some consistency of measurement within the acquired capability research, some of the studies within this review used partial measures. When items are selected and omitted from reliable and valid measures, what is purported to be measured may not actually be being measured, thus restricting interpretations and conclusions that can be drawn across studies [149]. Like single contributor studies and potential publication bias, partial use of measures is likely to hamper further understanding about suicide capability given that theory development is dependent on findings from psychometrically sound measures [150]. Third, most studies lacked diversity (e.g., racial/ethnicity) as the majority were U.S. participants that identified as Caucasian. This potentially distorts understandings of suicide capability. For example, methods used for suicide in the U.S. are different to other countries identified in this review, the most common method of suicide in Australia is hanging [151] whereas firearms are the most common method in the U.S. [54].

## Future research recommendations

Based on the limitations of the literature identified in this review, the following recommendations are offered to move our understanding of suicide capability forward. Studies that include multiple contributors are suggested to better reflect the ideation-to-action framework foundation and are more likely to assist in advancing suicide capability as a multifaceted concept. Given there is a dearth of research that has combined contributors, perhaps it is worthwhile to explore more than one contributor at a time using qualitative research grounded in lived experience [42, 43, 65, 152, 153]. This will help towards discovering insights about the nature and development of suicide capability because knowledge will be generated from individuals who evidently have a capability for suicide. Besides helping understand the "how" of suicide capability development, qualitative research helps generate ideas for quantitative research [154]. The ideas potentially generated from qualitative research can strategically guide further expansion of suicide capability research to build knowledge incrementally rather than reinforce and reproduce the piecemeal approach that appears to currently characterise the field.

Given the above, quantitative studies should replicate and expand on the various painful and provocative events identified in this review to consolidate and generate new knowledge about suicide capability. For example, the ACSS-FAD [61] could be added to painful and provocative event studies to refine the acquired contributor, or other measures that reflect additional contributors could be added to acquired capability studies. Further, given that suicide capability has been found to fluctuate daily [155], longitudinal designs are recommended as they can detect potential changes of suicide capability that cross-sectional designs are unable to identify. Ecological momentary assessment studies are one such approach that may help further elucidate the dynamic nature of suicide capability [156].

The current mixture of measures being used hampers development of the field. Measurement issues with single-item assessments of suicide attempts [157] or contributors of suicide capability [66] are recurring issues. As Kramer et al. [158] previously argued with regards to acquired capability measures, future research will benefit from using psychometrically sound measures rather than partial versions to allow accurate interpretations and conclusions to be drawn across studies. Moreover, using the same scales is necessary for replication studies and continuity for theory development [159]. It is acknowledged that participant burden and ethical considerations need to be considered when selecting measures [160] and this may account for partial version use.

Given the lack of diversity in most studies, it is suggested that exploring suicide capability among more diverse populations is warranted. There may be painful and provocative events

specific to minority populations (e.g., discrimination) that contribute to capability which are not yet considered within theoretical models. Further, additional research across the world is required to develop understanding about potential geographic contributors beyond firearms in the U.S.

## Protocol modifications

Given that scoping reviews are iterative in nature, modifications were made to the protocol throughout the screening process as a result of discussions between the two reviewers. During the screening stage it became apparent that many studies did not explicitly define a suicide attempt as per Silverman et al. [30]. However, given that studies used definitions that were variations of the Silverman et al. definition and indicated intent, or whereby the participant reported that they intended to die from their suicide attempt, these studies were included. Other studies were excluded as they did not provide evidence about contributing factors of a suicide attempt, such as prevalence of suicide rates. Studies that did not distinguish suicide attempters from suicide ideators were excluded because it is necessary for the groups to be separated in analyses to ascertain whether factors are contributing to an attempt or ideation or both. This was inadvertently overlooked when designing the protocol. Some studies did not report the age range of samples, and some comprised university students which included participants aged younger than 18 years. For these studies to be included, we deferred to the mean age at or above 18 years.

## Review strengths and limitations

Although this review has provided a map of the suicide capability literature, it is not without its limitations. First, this review excluded 17 identified studies that were published in a language other than English. Despite requests for translated versions from the corresponding authors of these studies, no English full-text translations were available or provided. This exclusion may have missed important culture-specific factors and the findings from these studies may provide evidence that either contradicts or supports the findings from the review. Second, 25 studies were excluded as participants were under the age of 18 years. There may be developmental (e.g., coping skills [28]) and motivational (e.g., interpersonal problems [29]) aspects that could potentially be associated with adolescents and children's capability for suicide that this review did not capture. Therefore, future reviews may choose to focus on this age group. Third, experts were not consulted to discuss preliminary findings. However, peer-review feedback from an earlier version of this review helped refine the review premise and search strategy which facilitated the clarification of previously ambiguous findings.

A strength of this review was working to an a priori peer reviewed published protocol [24]. Rationale for modifications from the protocol has been provided, including the addition of a stage that was the amended and re-run search strategy to capture additional studies ($n = 33$) that brought the review up to date. Almost half of the studies ($n = 43$) in this review were published in the past five years (i.e., since 2017) indicating this review was timely and therefore needed. The rationale for the modifications allowed for potential biases to be identified and considered [161]. Further, an independent random audit quality check of the data extraction was completed to ensure rigour of synthesis. Another strength was the large number of databases and grey literature sources included in this review compared to previous reviews (see [6, 7, 18–21]). This was important for capturing literature that may have been overlooked by these reviews. Unlike narrative reviews, the transparency of this scoping review was a strength because of the clear and careful documentation of the process, the independent searching and

screening, the random audit of data extraction templates, and using the PRISMA-ScR [162] to highlight methodological rigour in S2 Table.

## Conclusion

Suicide capability has been posited as a key concept that facilitates our understanding of why some individuals act on suicidal thoughts while most do not. The findings of this review suggest painful and provocative events provide most clarity in understanding this movement, from an acquired capability perspective, and thus capture some of capability's connection with suicidal behaviour. There are additional emerging areas of promise (e.g., cognitive contributors), however further research is needed to determine if they are contributors of suicide capability and how an individual develops this capability. The movement from ideation-to-action is complex [5–9, 15–17], yet the focus of most studies reduces this movement to a single factor isolated in time, which makes it difficult to see how such research can meaningfully contribute to advancing theoretical understandings of suicide capability. Continuing to research suicide capability in an individual contributor way utilising cross-sectional study designs potentially prevents generation of new knowledge that can be used to better understand the movement from ideation-to-action and save lives. Therefore, research that utilises a combination of contributors is needed to explicate the potential dynamic interplay of contributors and lead to an increased understanding of suicide capability. This review has publicised the current state of the field and it is hoped that it has provided an evidential platform for future research to strategically enhance our theoretical understandings of suicide capability.

## Supporting information

**S1 Table. Study aims and theoretical relationships.** This table provides the aim of each article and how findings are linked to the concept of suicide capability, where applicable.
(DOCX)

**S2 Table. Preferred reporting items for systematic reviews and meta-analyses extension for scoping reviews (PRISMA-ScR) checklist.**
(DOCX)

## Acknowledgments

The authors would like to thank Philip Batterham for critically reading the manuscript and providing insightful comments that helped strengthen the review. We would also like to thank the anonymous reviewers for suggestions that guided the addition of the sixth stage in the review process.

## Author Contributions

**Conceptualization:** Luke T. Bayliss.

**Formal analysis:** Luke T. Bayliss.

**Investigation:** Luke T. Bayliss, Steven Christensen.

**Methodology:** Luke T. Bayliss, Andrea Lamont-Mills, Carol du Plessis.

**Writing – original draft:** Luke T. Bayliss.

**Writing – review & editing:** Luke T. Bayliss, Steven Christensen, Andrea Lamont-Mills, Carol du Plessis.

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
