## [Decision Letter · Decision Letter 0]

28 Jul 2022

PONE-D-22-12018Suicide capability within the ideation-to-action framework: A systematic scoping reviewPLOS ONE

Dear Dr. Bayliss,

Thank you for submitting your manuscript to PLOS ONE. After careful consideration, we feel that it has merit but does not fully meet PLOS ONE’s publication criteria as it currently stands. Therefore, we invite you to submit a revised version of the manuscript that addresses the points raised during the review process. Can you please carefully address the reviewers' concerns below?

We look forward to receiving your revised manuscript.

Kind regards,

Avanti Dey, PhD

Staff Editor

PLOS ONE

Journal Requirements:

Reviewers' comments:

Reviewer's Responses to Questions

**Comments to the Author**

1. Is the manuscript technically sound, and do the data support the conclusions?

Reviewer #1: Partly

Reviewer #2: Yes

2. Has the statistical analysis been performed appropriately and rigorously? 

Reviewer #1: N/A

Reviewer #2: Yes

3. Have the authors made all data underlying the findings in their manuscript fully available?

Reviewer #1: Yes

Reviewer #2: Yes

4. Is the manuscript presented in an intelligible fashion and written in standard English?

Reviewer #1: Yes

Reviewer #2: Yes

5. Review Comments to the Author

Reviewer #1: While this manuscript addresses relevant points in suicide research it would profit from improvements at two points, which both share the same theme: A clearer discrimination between scoping reviews and other forms of review (i.e. systematic reviews). This is of importance as not all readers are aware of the difference between these types of reviews.

Firstly, in the in introduction section authors argued for the relevancy of their scoping review, mostly comparing it to other forms of reviews. But, as these other reviews were often systematic reviews, I argue that no equivalency exists between the scoping review presented here and the systematic reviews referenced. For example to call a systematic review "to narrow" (page 5 line 101) in comparison to a scoping review is a non-equivalent comparison, as both have different aims.

The paper would be stronger and more accepted by the broader scientific community, if the authors would refrain from attacking these other forms of reviews.

The fact that no (recent) scoping review exists to this research question and why it should exist, would be more than enough to argue for publication. An argument which was well executed by the authors, but is muddled by the imprecise use of the term review, if each referenced review would be specified within the paper (e.g. systematic review, practitioners review…) we would all get a clearer picture of why this scoping review is needed and how it adds to the broader research field.

Secondly, in the discussion section authors argue that the field does not reflect the multifaceted nature of the research question and that most studies used single factors.

I think this conclusion is to hash.

As a major limitation of scoping reviews is the large amount of potential oversight inherent in these very broad searches. Using the FAD as an example, authors stated they found four studies (Line 396). My (systematic) overview of the FAD papers, all including suicide ideation and suicide attempts, has around thirty entries, of which maybe five to ten are of adolescent samples.

This discrepancy in numbers shows the inherent limitation of scoping reviews. As scoping reviews use very general search strings they tend to over-proportionally miss multivariate studies, as the naming of multivariate studies often do not directly reference all used variables.

I therefore find the conclusion of the authors a bit harsh regarding to few multivariate approaches, given that scoping reviews have an inherent tendency to overlook these kinds of studies.

This weakness should either be addressed the limitation or discussion section. A potential way to limited the impact of such oversights, or strengthen it, would be to use the systematic reviews cited in the introduction section, to fill up found studies and contextualise study results.

Reviewer #2: Overall the paper is excellent. Authors do a great job of introducing rationale, address a needed gap in the literature, sound methodology, and I thought authors including a demographic table was a particular strength of this review. The approach of considering dispositional and practical capability factors, and finding of cognitive factors, are also particularly novel additions to the conceptualization of suicide capability. The paper has been written in an excellent intelligible fashion. This review comes with a few minor suggestions:

1) I suggest authors expand on the definitions/contributions of dispositional and practical factors to the suicide capability model on page 4 of the introduction, in a manner similar to how authors introduce acquired capability for suicide.

2) Given the lack of diversity (racial/ethnic, gender, SES) in majority of the articles discussed in the review, this should be added as a limitation. Further, exploring these constructs among more diverse populations that consider events that are shown to be painful and provocative for minority populations (e.g., discrimination) needs to be considered as an area for future research.

6. PLOS authors have the option to publish the peer review history of their article (what does this mean?). If published, this will include your full peer review and any attached files.

Reviewer #1: **Yes: **Jim Schmeckenbecher

Reviewer #2: No

---

## [Author Response · Author response to Decision Letter 0]

3 Aug 2022

Please see our responses in the Response to Reviewers file.

---

## [Decision Letter · Decision Letter 1]

31 Aug 2022

PONE-D-22-12018R1Suicide capability within the ideation-to-action framework: A systematic scoping reviewPLOS ONE

Dear Dr. Bayliss,

Thank you for submitting your manuscript to PLOS ONE. After careful consideration, we feel that it has merit but does not fully meet PLOS ONE’s publication criteria as it currently stands. Therefore, we invite you to submit a revised version of the manuscript that addresses the points raised during the review process.

As you know, PLOS requires at least two reviewers, therfore in spite of already revising your paper, now you have new comments. I kindly ask you therefore to address the comments made by reviewer 3 before we can proceed to make a final decision on your work.

We look forward to receiving your revised manuscript.

Kind regards,

Xenia Gonda

Academic Editor

PLOS ONE

Reviewers' comments:

Reviewer's Responses to Questions

**Comments to the Author**

1. If the authors have adequately addressed your comments raised in a previous round of review and you feel that this manuscript is now acceptable for publication, you may indicate that here to bypass the “Comments to the Author” section, enter your conflict of interest statement in the “Confidential to Editor” section, and submit your "Accept" recommendation.

Reviewer #1: All comments have been addressed

Reviewer #3: All comments have been addressed

2. Is the manuscript technically sound, and do the data support the conclusions?

Reviewer #1: Yes

Reviewer #3: Yes

3. Has the statistical analysis been performed appropriately and rigorously? 

Reviewer #1: Yes

Reviewer #3: N/A

4. Have the authors made all data underlying the findings in their manuscript fully available?

Reviewer #1: Yes

Reviewer #3: Yes

5. Is the manuscript presented in an intelligible fashion and written in standard English?

Reviewer #1: Yes

Reviewer #3: Yes

6. Review Comments to the Author

Reviewer #1: The authors addressed previously stated concerns.

All other remaining differences are due to a difference in opinion about what is most relevant. As these do not extend to research quality, rigorosity or relevance of the paper, publication is recommended.

Reviewer #3: This is a A scoping review of suicide capability studies published between January 2005 to January 2022. The topic is interesting but the ms needs revisions before being considered for publication. For example formatting and position of the tables should be reconsidered. As it is now the ms is very long and sometimes difficult to follow.

Introduction

page 101 line 82 and followings. Please give some information on the acquired factors.

"The templated" check whether correct

Results

"(see Fig 1)" I think that "Figure" shlould be included and not "Fig"

Page 124 Please check the sentence "Most studies focused on individuals without a psychiatric"

page 145 Please check the sentence "was found to be related with suicide attempt history [118] more so than individuals who had"

Please check the sentence "Three association studies found cognitive difficulties were reported"

Discussion

PLease check "ambiguity about how these two constructs potentially effect acquired capability."

7. PLOS authors have the option to publish the peer review history of their article (what does this mean?). If published, this will include your full peer review and any attached files.

Reviewer #1: **Yes: **Jim Schmeckenbecher

Reviewer #3: No

---

## [Author Response · Author response to Decision Letter 1]

31 Aug 2022

We amended sentences based on Reviewer 3 comments and shifted tables 3 and 4 to increase readability whilst adhering to the journal’s formatting guidelines. Please see Response to Reviewers document for specific page and line numbers where changes were made.

---

## [Editor Report · Decision Letter 2]

28 Sep 2022

Suicide capability within the ideation-to-action framework: A systematic scoping review

PONE-D-22-12018R2

Dear Dr. Bayliss,

We’re pleased to inform you that your manuscript has been judged scientifically suitable for publication and will be formally accepted for publication once it meets all outstanding technical requirements.

Kind regards,

Xenia Gonda

Academic Editor

PLOS ONE
---

## [Editor Report · Acceptance letter]

5 Oct 2022

PONE-D-22-12018R2 

Suicide capability within the ideation-to-action framework: A systematic scoping review 

Dear Dr. Bayliss:

I'm pleased to inform you that your manuscript has been deemed suitable for publication in PLOS ONE. Congratulations! Your manuscript is now with our production department. 

Kind regards, 

on behalf of

Dr. Xenia Gonda 

Academic Editor

PLOS ONE